# Nitro-oleic acid enhances mitochondrial metabolism and ameliorates heart failure with preserved ejection fraction in mice

Marion Müller [1,2], Torben Schubert[1,2], Cornelius Welke [1,2], Tibor Maske [1,2], Thomas Patschkowski[3], Elfi Donhauser[1,2], Jacqueline Heinen-Weiler[4], Felix-Levin Hormann [5], Sven Heiles [5,6], Tina Johanna Schulz[1,2], Luisa Andrea Lengenfelder[1,2], Lucia Landwehrjohann[1,2], Elisa Theres Vogt[7], Bernd Stratmann [7], Jurek Hense [8], Simon Lüdtke[8], Martina Düfer [8], Elena Tolstik[5], Johann Dierks[5], Kristina Lorenz [5,9], Tamino Huxohl [10], Jan-Christian Reil[1], Vasco Sequeira [11], Francisco Jose Schopfer [12], Bruce A. Freeman [12], Volker Rudolph [1,2], Uwe Schlomann[1,2,13] & Anna Klinke [1,2,13] ✉

The prevalence of heart failure with preserved ejection fraction (HFpEF) is increasing, while treatment options are inadequate. Hypertension and obesity-related metabolic dysfunction contribute to HFpEF. Nitro-oleic acid ($NO_2$-OA) impacts metabolic syndromes by improving glucose tolerance and adipocyte function. Here we show that treatment with $NO_2$-OA ameliorates diastolic dysfunction and heart failure symptoms in a HFpEF mouse model induced by high-fat diet and inhibition of the endothelial nitric oxide synthase. Proteomic analysis of left ventricular tissue reveals that one-third of identified proteins, predominantly mitochondrial, are upregulated in hearts of $NO_2$-OA-treated HFpEF mice compared to naïve and vehicle-treated HFpEF mice. Increased mitochondrial mass and numbers, and enhanced mitochondrial respiration are linked with this response, as assessed by transmission electron microscopy and high-resolution respirometry. Activation of the 5'-adenosine-monophosphate-activated-protein-kinase (AMPK) signaling pathway mediates the enhancement of mitochondrial dynamics in hearts of $NO_2$-OA-treated HFpEF mice. These findings suggest that targeting mitochondrial function with $NO_2$-OA may represent a promising therapeutic strategy for HFpEF.

Heart failure with preserved ejection fraction (HFpEF) is a disorder with high morbidity and mortality that is increasing in prevalence in the aging populations of Western countries[1]. Given the heterogeneity and multifactorial nature of this syndrome, the definition of distinct phenogroups has been recently established to better stratify HFpEF patients[2-4]. The group of patients with cardiometabolic HFpEF exhibits obesity and metabolic syndrome. Systemic vascular inflammation and hypertension are further common phenomena of this phenogroup.

Sodium-glucose cotransporter 2 (SGLT2) inhibitors reduce the risk of cardiovascular death in HFpEF[5,6], with mechanisms of action remaining elusive. Apart from that, pharmacological options that directly target cardiac malfunction are not available, given that even in a particular phenogroup, the pathogenesis of diastolic dysfunction is multifactorial and incompletely understood. Pathophysiological hallmarks of HFpEF include metabolic remodeling of the myocardium, insulin resistance, and altered lipid handling. This is mediated by elevated

lipid levels, which promote a decrease in glucose uptake and consequent hyperinsulinemia[7]. This, together with additional risk factors such as physical inactivity, stimulates compensatory cardiomyocyte hypertrophy and causes microvascular dysfunction, resulting in reduced oxygen supply to the cardiomyocyte. Oxygen is required during aerobic energy generation, with fatty acid oxidation (FAO) consuming more oxygen than glucose oxidation (GO) or glycolysis. At the same time, FAO is the most effective way to generate adenosine triphosphate (ATP) per molecule substrate, with the amount of released energy being proportional to acyl chain length. The metabolic dysregulation of HFpEF impairs metabolic flexibility, so that mitochondrial metabolism cannot adapt to the variable cardiomyocyte demand for energy and oxygen. Given that insulin resistance shifts the cardiomyocyte substrate metabolism further towards FAO, the role of FAO in cardiometabolic HFpEF remains controversial[8,9]. In fact, lipotoxicity has been linked to the HFpEF pathophysiology. This is characterized by cardiomyocyte accumulation of lipid intermediates such as ceramide and diacylglycerol (DAG) due to the increased uptake of fatty acid (FA) and or a decreased rate of FAO[8,10]. Importantly, alterations in the lipid composition of cell and organelle membranes can disturb cellular integrity and also lead to mitochondrial dysfunction[11,12]. Critical mediators of metabolic regulation in cardiomyocytes include AMP-activated protein kinase (AMPK) and the deacetylase sirtuin 1 (SIRT1), which promote catabolic pathways, in particular FAO, in part via activation of mitochondrial biogenesis and the peroxisome-proliferator activated receptor (PPAR)-α.

Nitro-fatty acids are electrophilic byproducts of reactions of the nitrogen dioxide derived from the oxidation of NO and nitrite ($NO_2^-$) formed during metabolic and inflammatory responses with FA containing conjugated double bonds. Both endogenous and synthetic electrophilic small molecule nitroalkenes primarily react with a limited population of highly reactive and functionally significant cysteine moieties in enzymes and transcriptional regulatory factors, in turn modulating protein catalytic function, gene expression and overall promoting salutary adaptive cell signaling, metabolic and inflammatory responses. Endogenous plasma levels of non-esterified nitro-fatty acids under basal conditions are ~1–3 nM, and increase during inflammation and through dietary interventions to reach levels of 7–10 nM[13]. Similar levels are achieved in both animal models and after dosing human subjects with 150 mg $NO_2$-OA/day PO[14,15]. Notably, the free acid levels reflected by these plasma measurements do not include the esterified pool of nitro-fatty acids, which is ~20–30 fold greater than the free acid levels in plasma[16]. Experimental studies have explored the effects of the synthetic isomers (*E*)-9-and 10-nitrooctadec-9-enoic acid (nitro-oleic acid, $NO_2$-OA), nitrooctadeca-9,12-dienoic acid (nitro-linoleic acid, $NO_2$-LA) and the predominant endogenous small molecule nitroalkene isomers, 9-$NO_2$-CLA (9-nitrooctadeca-9,11-dienoic acid) and 12-$NO_2$-CLA (12-nitrooctadeca-9,11-dienoic acid), commonly referred as nitroconjugated linoleic acid ($NO_2$-CLA)[17–19]. An ongoing phase 2 clinical trial is evaluating the anti-inflammatory, tissue metabolic, microbiome and pulmonary function actions of (*E*)-10-nitrooctadec-9-enoic acid on airway dysfunction in obese asthmatics (NCT03762395). Small molecule nitroalkenes exert anti-inflammatory and adaptive repair actions in part via activation of the nuclear factor erythroid 2-related factor 2 (Nrf2) and inhibition of the nuclear factor 'kappa-light-chain-enhancer' of activated B-cells (NF-κB)[20]. Another impact of nitroalkene modulation of the reactive cysteine proteome is an improvement in energy metabolism, not only by limiting the pro-inflammatory milieu in liver and adipose tissue[21,22], but also by ligand-mediated activation of PPARs[23]. In leptin-deficient mice, obesity, glucose intolerance, and hyperglycemia were reduced by subcutaneous $NO_2$-OA administration[24]. In a high-fat diet (HFD) mouse model, right ventricular function was partly improved by $NO_2$-OA reduction of pulmonary arterial remodeling[21].

The two-hit HFpEF mouse model of combined HFD with $N_\omega$-nitro-L-arginine methyl ester hydrochloride (L-NAME) administration to inhibit the endothelial nitric oxide synthase (eNOS), established by Schiattarella and colleagues[25], unites the two major HFpEF risk factors - hypertension and metabolic syndrome. We employed this 15-week HFD + L-NAME model to study LV function responses to $NO_2$-OA administered in the last 4 weeks of treatment.

## Results

### Nitro-oleic acid mitigates heart failure in mice with HFpEF

A mouse model of HFpEF induced by a combination of HFD and inhibition of eNOS with L-NAME for 5 or 15 weeks (wk) has been described and extensively characterized before by the group of Joseph Hill[25–28]. We chose the 15 wk HFD + L-NAME model and, to gain perspective on potential limitation or reversal of disease phenotype by $NO_2$-OA, we treated mice with $NO_2$-OA (HFD + L-NAME $NO_2$-OA), non-electrophilic oleic acid (HFD + L-NAME OA) or vehicle (HFD + L-NAME vehicle) during wk 12 to 15 via osmotic pumps (Figs. 1a, S1a).

After 11 wk, HFD + L-NAME mice developed an HFpEF phenotype reflected by preserved LV ejection fraction (LVEF), diastolic dysfunction, LV hypertrophy (Fig. S1b) and interstitial fibrosis (Fig. S2a). Remarkably, treatment with $NO_2$-OA significantly reduced the ratio of early diastolic mitral inflow velocity to early diastolic mitral annulus velocity (E/e' ratio) compared to vehicle-treated HFD + L-NAME mice (Fig. 1c, d). The normalization of the early diastolic mitral annulus velocity (e') after treatment with $NO_2$-OA was primarily responsible for the improved diastolic function (Fig. S3a), whereas the transmitral profile (E/A ratio) was not altered among the groups (Fig. S3b). Accordingly, the isolvolumetric relaxation time (IVRT) was prolonged by HFD + L-NAME and normalized by $NO_2$-OA-treatment (Fig. 1d). Importantly, the therapeutic effect of $NO_2$-OA also became evident by a restoration of LV global longitudinal strain (Fig. 1e). The E/e' ratio related to end-diastolic volume reflecting LV compliance was normalized by $NO_2$-OA treatment as well (Fig. 1f). Corroborating the profound impairment of compliance, invasive hemodynamic measurements revealed enhanced LV stiffness in HFD + L-NAME mice, which was blunted by $NO_2$-OA (Fig. 1h). Consequently, LV end-diastolic pressure (LVEDP) was significantly increased in vehicle-treated but not in $NO_2$-OA-treated HFD + L-NAME mice (Fig. 1g). Importantly, treatment with non-electrophilic oleic acid (HFD + L-NAME OA) had no significant effect (Fig. S1), affirming the key impact of the nitroalkene group, which confers electrophilicity and promotes the instigation of systemic responses by post-translational modification of predominantly target cysteine residues. Data shown in Suppl. Fig. S1 were derived from an initial cohort of mice, and data shown in Fig. 1 represent a second independent cohort of mice, which confirmed the echocardiography results from the intitial cohort.

To test whether disturbed LV diastolic function resulted in heart failure symptoms, exercise capacity of mice was assessed by treadmill performance. Vehicle-treated HFD + L-NAME mice achieved a significantly lower work compared to control mice, whereas $NO_2$-OA-treated mice showed only subtle impairments (Fig. 1i). Another established marker of disease severity in heart failure and an important predictor of adverse outcome in patients is the left atrial (LA) area, which was markedly increased in vehicle-treated and normalized to control level in $NO_2$-OA-treated HFD + L-NAME mice (Fig. 1j). Given that LA enlargement is associated with pulmonary hypertension (PH), pulmonary artery muscularization, an indicator of PH, was analyzed in lung sections. Smooth muscle cell area of pulmonary arterioles was significantly greater in vehicle- but not in $NO_2$-OA-treated HFD + L-NAME mice compared to control animals (Fig. 1k). Despite these profound effects of $NO_2$-OA on the HFpEF phenotype, LV hypertrophy and interstitial fibrosis were not significantly influenced by $NO_2$-OA (Figs. 1b, S2). Systemic inflammation, as reflected by increased plasma levels of the leukocyte enzyme

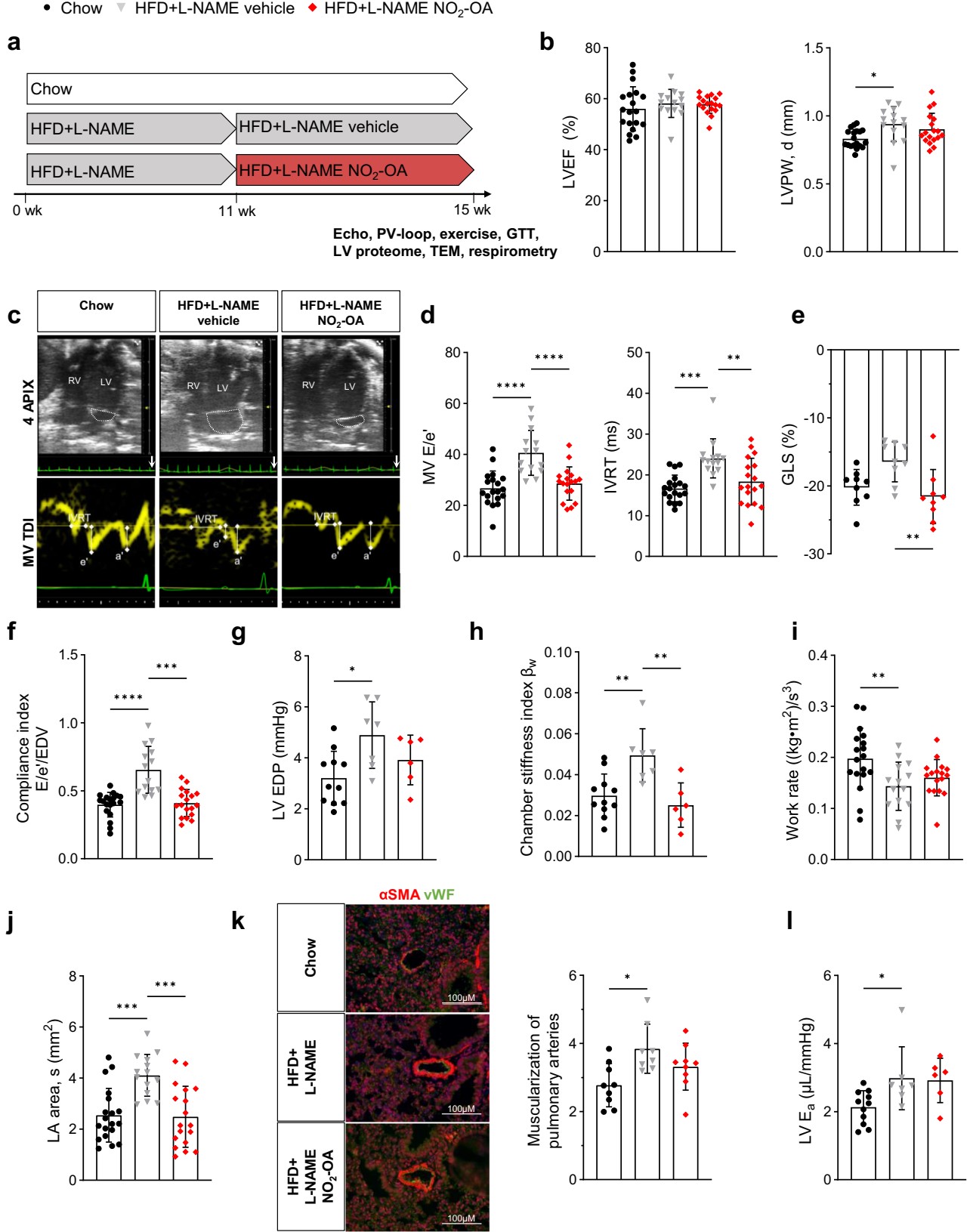

myeloperoxidase in HFD + L-NAME mice, was also not altered by NO$_2$-OA (Fig. S4a). In line with that, systemic vascular resistance and vascular nitric oxide bioavailability were increased in both HFD + L-NAME groups compared to chow mice (Figs. 1l, S4b). In contrast to the systemic vasculature, biomarkers of inflammation, i.e., oxidative modifications, did not become evident in LV

myocardium (Fig. S4c), which might be due to the compensatory upregulation of anti-oxidant enzymes in vehicle-treated HFD + L-NAME mice (Fig. S4d).

To consider potential sex differences, the efficiency of NO$_2$-OA was analyzed in female HFD + L-NAME mice by echocardiography (Fig. S5). The HFpEF phenotype was less pronounced in female mice

**Fig. 1 | Nitro-oleic acid mitigates heart failure in mice with HFpEF.** Mice received a high-fat diet (HFD) and the endothelial nitric oxide synthase inhibitor L-NAME (HFD + L-NAME) or a standard chow diet (Chow) for 15 weeks (wk). They were treated with vehicle or nitro-oleic acid ($NO_2$-OA) during the last 4 wk. Echocardiography (Echo), glucose tolerance test (GTT), hemodynamics (PV-loop), exercise, transmission electron microscopy (TEM), proteomics of left ventricular tissue (LV proteome) and respirometry of isolated cardiomyocytes was conducted at the final time point. **a** Experimental protocol. **b** Left ventricular ejection fraction (LVEF) and left ventricular posterior wall thickness (LVPW, d) ($N = 19/14/18$). **c** Representative ultrasound images of 4-chamber B-Mode (upper panel) and LV tissue doppler at mitral annulus (lower panel) to assess diastolic function. **d** Ratio of early diastolic mitral inflow velocity to early diastolic mitral annulus velocity (MV E/e´) and isovolumetric relaxation time (IVRT) ($N = 19/14/18$). **e** LV global longitudinal strain (GLS) ($N = 9/9/9$). **f** Compliance index shown as E/e´ in ratio to the individual end diastolic volume (EDV) ($N = 19/14/18$). **g, h** LV end-diastolic pressure (LV EDP) (**g**) and chamber stiffness index ß$_W$ (**h**) derived from hemodynamics mesurements ($N = 11/7/6$). **i** Work rate calculated from maximum exhaustion exercise test ($N = 18/14/17$). **j** Left atrial area (LA area) ($N = 19/14/18$). **k** Representative images of muscularization of pulmonary arteries by staining α-smooth muscle actin (αSMA) and von Willebrand factor (vWF) (left) and quantification of αSMA-positive area related to vessel circumference of arterioles (right) ($N = 9/8/9$). **l** LV arterial elastance (LV E$_a$) ($N = 11/7/6$). Data are presented as mean ± standard deviation. Statistical significance was calculated by One-way ANOVA followed by Bonferroni´s post-hoc test for **b, d, e, g-l**. Statistical significance for **f** was calculated by Kruskal-Wallis-test followed by Dunn´s multiple comparison test. Only statistically significant differences are indicated. $p$*<0.05, **<0.01, ***<0.001, ****<0.0001.

after receiving HFD + L-NAME for 11 wk compared to male HFD + L-NAME animals, reflected by significantly increased LV hypertrophy and LA area, but no change in E/e' ratio (Fig. S5a). However, following the data obtained in male HFD + L-NAME mice, administration of $NO_2$-OA for 4 wk significantly improved LA area and diastolic function (E/e') also in female HFD + L-NAME mice (Fig. S5b), affirming the efficacy of $NO_2$-OA in metabolic cardiomyopathy in both sexes.

Taken together, treatment with $NO_2$-OA eliminated diastolic dysfunction and mitigated heart failure in HFD + L-NAME mice without significantly influencing structural remodeling and inflammation.

## Nitro-oleic acid restores protein profiles in LV of mice with HFpEF

To define the effects of $NO_2$-OA treatment on the LV proteome, LV tissue from control, HFD + L-NAME vehicle and HFD + L-NAME $NO_2$-OA mice were analyzed by LC-MS. In total, 1,878 proteins were identified. Unexpectedly, principal component analysis indicated relatively slight differences in protein abundances between HFD + L-NAME and control mice, while HFD + L-NAME $NO_2$-OA mice showed distinct differences in protein abundance (Fig. 2a). In LV of HFD + L-NAME mice, only 1.4% of total proteins had significantly higher abundances, while 1.2% of total proteins had lower levels compared to control (Fig. 2b). Due to the increased lipid load in vehicle-treated HFD + L-NAME mice, KEGG pathway analysis identified increased protein levels in cholesterol and lipid metabolism as well as redox signaling and lipid homeostasis in peroxisome in LV of HFD + L-NAME compared to control mice (Fig. 2c). Comparison of proteome data of $NO_2$-OA- versus vehicle-treated HFD + L-NAME mice showed, that 2.7% of total proteins were expressed at significantly lower levels, but 26.3% of total proteins had significantly increased abundance compared to vehicle (Fig. 2d). In addition, LC-MS analysis showed that 68% of proteins (15 proteins) with decreased abundance in HFD + L-NAME compared to control hearts had a significantly higher abundance after $NO_2$-OA compared to vehicle treatment (Fig. 2e). Thus, treatment with $NO_2$-OA significantly normalized protein abundance after 15 wk of HFD + L-NAME administration. Further analyses of these protein responses and their impact is presented in greater detail below.

## Nitro-oleic acid improves impaired glucose metabolism in mice with HFpEF

As expected, body weight and plasma glucose levels were markedly increased in HFD + L-NAME compared to mice fed with standard chow (Fig. 2f). As described for obese leptin-deficient mice[24], treatment with $NO_2$-OA significantly normalized glucose tolerance compared to vehicle. In addition, body weight was reduced (Fig. 2f). In line with the metabolic phenotype, analysis of isolated pancreatic islets revealed a rise in insulin content, indicative of compensatory islet enlargement in response to HFD + L-NAME. Furthermore, β-cell size was significantly increased. Similar trends were observed for islet- and α-cell size. Administration of $NO_2$-OA normalized these parameters (Fig. S6). The profile of glucose handling proteins in LV tissue was significantly impacted by HFD + L-NAME, as reflected by a heatmap showing changes in proteins associated with glycolysis/gluconeogenesis (Fig. S7). Related to this, protein abundance of the glucose transporter (GLUT4) was significantly decreased, and the pyruvate dehydrogenase kinase (PDK4) was significantly increased in HFD + L-NAME mice compared to control mice, both of which were markedly attenuated by $NO_2$-OA treatment (Fig. 2g, h). Consequently, enhanced phosphorylation of pyruvate dehydrogenase (PDH) was detected in HFD + L-NAME mice treated with vehicle, indicating an attenuation of GO rates, which was much less pronounced in $NO_2$-OA-treated animals (Fig. 2h). Given that PDK4 and PDH are expressed solely in mitochondria, their protein levels were normalized to protein levels of COX4, representative for mitochondrial amount in heart homogenates. The findings point out, that $NO_2$-OA ameliorates disturbed glucose metabolism.

## Nitro-oleic acid increases mitochondrial protein level in LV of mice with HFpEF

LC-MS proteomic analysis revealed 568 out of 1,139 mitochondrial proteins annotated in the mitoCarta database[29] being detected. Remarkably, in LV of $NO_2$-OA-treated mice, 24.1% of these mitochondrial proteins were significantly more abundant than in vehicle-treated mice, which is impressively demonstrated by the heatmap (Fig. 3a). MitoCarta localization[29] illustrates the functional affiliation of these mitochondrial proteins and the percentage of those significantly increased by $NO_2$-OA treatment (Fig. 3b). This analysis revealed that mitochondrial outer membrane proteins are particularly enriched by $NO_2$-OA (Fig. 3b). Figure 3c illustrates five of the highest abundant mitochondrial proteins identified in $NO_2$-OA-treated HFD + L-NAME mouse hearts by LC-MS compared to vehicle-treated HFD + L-NAME mice, emphasizing a significant upregulation of these critical mitochondrial proteins by $NO_2$-OA. This was confirmed by immunoblot analysis of the mitochondrial proteins cytochrome c oxidase subunit 4 (COX4) and the mitochondrial transport protein TOM70 (Fig. 3d). Interestingly, from the 13 mitochondrially encoded proteins, 8 were detected by LC-MS and shown to be significantly enhanced after treatment with $NO_2$-OA in HFD + L-NAME compared to control and vehicle.

Ultrastructural analysis using transmission electron microscopy (TEM) revealed that mitochondrial architecture was markedly altered in both HFD + L-NAME groups compared to control mice (Fig. 3e). In addition to darkly stained mitochondria with intact cristae structure, early cristae fragmentation (white arrow), cristae adhesion (white asterisk), severely swollen and ruptured mitochondria (black asterisk) and an overall intact mitochondrial distribution were observed (Fig. 3e). The sarcomeric structure was regular. Increased cytosol between and around mitochondria was observed in both vehicle- and $NO_2$-OA-treated HFD + L-NAME mice (black arrows), which was particularly enriched with glycogen in vehicle-treated mice. Interestingly, morphometric analysis revealed a shift in individual mitochondrial size in $NO_2$-OA-treated mice, with these cardiomyocytes exhibiting significantly smaller mitochondria (shift towards a smaller bin center of

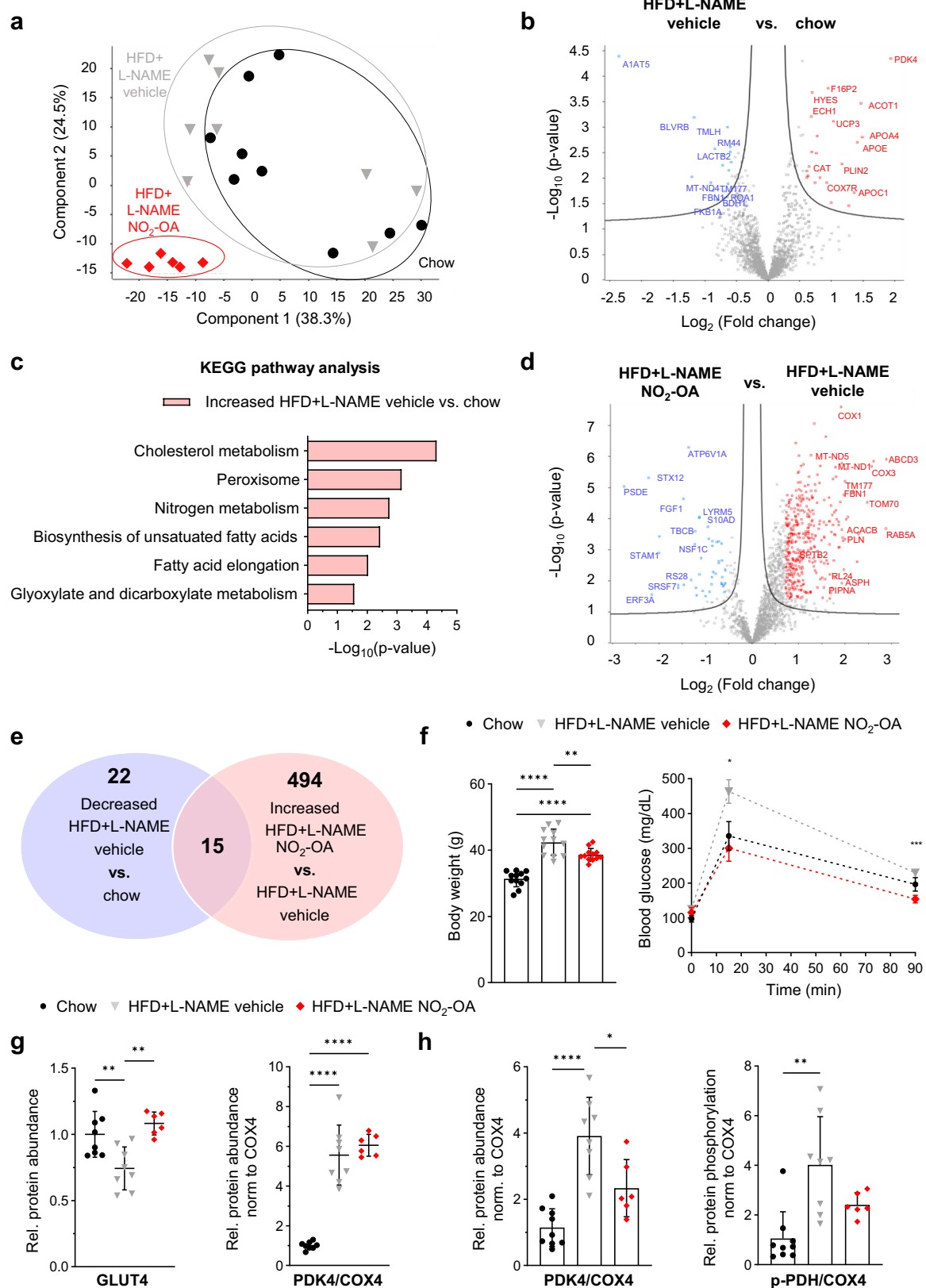

0.4 μm², Fig. S8), in concert with significantly increased numbers of mitochondria (Figs. 3f and S8). The observation of smaller but markedly increased numbers of mitochondria aligns with the increased abundance of outer mitochondrial membrane proteins detected by LC-MS. Interestingly, mitochondrial number data had a significant negative correlation with IVRT, indicating an association with

improved active LV relaxation (Fig. 3g). Of note, mitochondrial transcription factor A (TFAM) was significantly enhanced by NO₂-OA in HFD + L-NAME mice compared to chow and HFD + L-NAME mice (Fig. 3h), pointing towards increased mitochondrial biogenesis. In line, mitochondrial DNA copy number was significantly increased in these animals (Fig. 3h) and the relative number of mitochondrial DNA copies

**Fig. 2 | Nitro-oleic acid treatment alters myocardial proteome and glucose metabolism in HFpEF mice. a** Principal component analysis (Chow: $N = 9$; HFD + L-NAME vehicle: $N = 8$; HFD + L-NAME NO$_2$-OA: $N = 6$). Volcano-plots comparing high-fat diet (HFD) + L-NAME vehicle with chow mice (**b**) and NO$_2$-OA-treated with vehicle-treated HFD + L-NAME mice (**d**). Significantly changed proteins are marked (significance $\geq$ -log$_{10}$(0.05); red: fold change > log$_2$(1.5); blue: fold change < -log$_2$(1.5)). Protein names, log fold changes and $p$-values are listed in the source data file. **c** KEGG pathway enrichment analysis of significantly increased proteins of HFD + L-NAME vehicle compared to chow mice (protein count $\geq$ 2; significance $\geq$ -log$_{10}$(0.05)). **e** Venn diagram of proteins significantly decreased in the HFD + L-NAME vehicle compared to the chow group and proteins significantly increased in NO$_2$-OA-treated compared to vehicle-treated HFD + L-NAME mice. **f** Body weight ($N = 12/12/12$) and glucose tolerance test ($N = 7/7/7$). **g** Protein abundances of solute carrier family 2, member 4 (GLUT4) and pyruvate dehydrogenase kinase 4 (PDK4) relative to cytochrome c oxidase 4 (COX4) derived from liquid chromatography–mass spectrometry ($N = 8/8/6$). **h** Quantification of immunoblots (IB) for PDK4 relative to COX4 and phosphorylated pyruvate dehydrogenase (p-PDH) relative to COX4 ($N = 9/8/6$). Protein levels were normalized to the total protein amount assessed on IB by fluorescence labeling ($N = 9/8/6$) (Fig. S12 and S13). Data are presented as mean ± standard deviation for **f** (left), **g** and **h**. **f** (right) shows mean ± standard error of the mean. Data in **g** and **h** are shown relative to the mean of the chow group. Statistical significance for **f** (left), **g** and **h** (left) was calculated by One-way ANOVA followed by Bonferroni´s post-hoc test. Kruskal-Wallis-test followed by Dunn´s multiple comparison test was used for **h** (right). Two-way ANOVA followed by Bonferroni´s multiple comparisons test was used for **f** (right). * indicates differences between HFD + L-NAME vehicle and HFD + L-NAME NO$_2$-OA. Statistical significance was calculated with Perseus (MPI Biochemistry) for **b** and **d** and with two-tailed Fisher's exact test for **c**. Only statistically significant differences are indicated. $p$ *<0.05, **<0.01, ****<0.0001. Outliers in **g** were identified using ROUT method and excluded from analysis.

strongly correlated with the protein expression of TOM70 and COX4 (Fig. S9). Taken together, multiple lines of evidence reveal a profound expansion of mitochondrial numbers and density in NO$_2$-OA-treated mice, possibly due to enhanced mitochondrial biogenesis and/or dynamics.

### Nitro-oleic acid upregulates sirtuin expression via induction of AMPK signaling

KEGG pathway analysis of protein expression changes by NO$_2$-OA identified AMPK signaling, an important mediator of mitochondrial biogenesis, dynamics and metabolism, as among the most significantly enriched pathways (Fig. 4a). The LC-MS heatmap of proteins associated with AMPK signaling illustrates a significantly greater protein abundance after treatment with NO$_2$-OA in LV of HFD + L-NAME compared to chow and HFD + L-NAME mice (Fig. 4b). Immunoblot analysis revealed significantly greater AMPK phosphorylation at threonine-172 in NO$_2$-OA-treated compared to vehicle-treated HFD + L-NAME animals as well as compared to controls, whereas AMPK protein levels were not different between groups (Fig. 4c). The activation of AMPK significantly increased mRNA expression of the NAD$^+$-dependent deacetylase sirtuin 1 (*Sirt1*), as previously[30] (Fig. S10a). Also, SIRT1 protein levels were significantly enhanced by NO$_2$-OA in HFD + L-NAME mice (Fig. 4c). SIRT1 is located in the cell nucleus and mediates gene transcription by deacetylation of transcription factors, such as the master regulator of mitochondrial biogenesis, peroxisome proliferator-activated receptor gamma coactivator 1-alpha (PGC-1α)[31,32]. In line with this, an increase of the mitochondrial deacetylase sirtuin 3 (SIRT3) expression was detected after treatment with NO$_2$-OA, as compared to chow and HFD + L-NAME mice (Fig. 4c). Sirtuin activity might also be further induced by elevated NAD$^+$ levels after treatment with NO$_2$-OA, since there were significantly increased mRNA levels of nicotinamide nucleotide adenyl transferase 1 (*Nmnat1*) (Fig. S10b), an enzyme central to the biosynthesis of NAD$^+$. Taken together, NO$_2$-OA induces AMPK activation in LV myocardium in HFpEF.

### Nitro-oleic acid increases mitochondrial respiration

To analyse the effects of NO$_2$-OA on mitochondrial respiration, high-resolution respirometry of isolated permeabilized cardiomyocytes was performed using a standardized substrate-uncoupler-inhibitor titration protocol. Carnitine and palmitoyl-CoA were used to test for FA and carnitine palmitoyltransferase 1B (CPT1B)-dependent respiration, with pyruvate, glutamate, malate and succinate used as substrates for complexes I and II respiration. Both CPT1B-dependent and -independent oxidative phosphorylation (OXPHOS) were significantly higher in cardiomyocytes of NO$_2$-OA-, but not vehicle-treated HFpEF mice compared to control animals (Fig. 5a). Similar results for NO$_2$-OA were obtained from the analysis of isolated adult murine cardiomyocytes from healthy animals upon cultivation for 48 h under normal glucose conditions (Ctrl) and metabolic stress conditions induced by high glucose, endothelin-1 (ET-1) and hydrocortisone (HC), as previously[33] (Fig. 5b–e). Immunoblot and immunocytochemical analysis (Fig. 5c, d) revealed enhanced mitochondrial protein expression accompanied by modest activation of AMPK by NO$_2$-OA in these metabolically stressed hyperglycemic cardiomyocytes. Respirometry showed significantly increased OXPHOS after treatment with NO$_2$-OA (Fig. 5e).

Given the marked increase in FA-dependent respiration in NO$_2$-OA-treated HFpEF mice, lipid handling metabolism and trafficking protein responses were analyzed in the LV proteome response data (Fig. 6a). Several proteins responsible for FA metabolism, amongst them CPT1B (Fig. 6b), were much more abundant in NO$_2$-OA-treated mice, whereas only subtle changes were detected in LV of HFD + L-NAME mice compared with control mice (Fig. 6a). Together with a profound increase in protein amount of FA translocase (FAT, CD36) (Fig. 6b), this points towards an augmentation of cardiac FA uptake and metabolism by NO$_2$-OA. Indeed, the area of lipid droplets in isolated cardiomyocytes of chow and HFpEF mice was increased in vehicle-treated HFpEF mice and was significantly attenuated in the NO$_2$-OA treatment group (Fig. 6c). A lipidomics analysis of myocardial tissue revealed augmented levels of diacylglycerols, triacylglycerols and ceramides in both HFD + L-NAME groups compared to chow (Fig. S11). These data indicate that NO$_2$-OA induces catabolic processes and enhances FA intake and metabolism in cardiomyocytes, which might compensate for a myocardial energy deficit in HFpEF.

## Discussion

Our study reveals that a small molecule electrophile modulator of the cysteine proteome, NO$_2$-OA, significantly alleviated heart failure symptoms and improved diastolic function in a two-hit HFpEF mouse model. This effect was mediated by an alteration in mitochondrial function and protein levels in the LV of NO$_2$-OA-treated mice. The abundance of mitochondrial proteins, including those responsible for lipid handling and FAO, was significantly increased in LV tissue of NO$_2$-OA-treated mice. Mitochondria numbers were increased, and both OXPHOS and FAO were enhanced.

These pharmacologic actions were mediated by the activation of AMPK in HFD + L-NAME mice and isolated cardiomyocytes treated with NO$_2$-OA. AMPK is central to the regulation of cellular catabolic pathways, thereby inducing FAO and both mitochondrial biogenesis and homeostasis[34,35]. LC-MS and immunoblot analyses of LV tissues also revealed a significant increase in the abundance of AMPK-regulated proteins in NO$_2$-OA-treated mice. NO$_2$-OA actions, primarily a consequence of protein nitro-alkylation of nucleophilic cysteine[36], also activate the deacetylase SIRT6 upon NO$_2$-OA adduction[37]. Here, we found SIRT1 and SIRT3 protein levels also increased in NO$_2$-OA-treated mice, which are downstream targets of AMPK. Of note, SIRT1-induced activation of AMPK and positive feedback mechanisms between AMPK activation and SIRT1 induction as

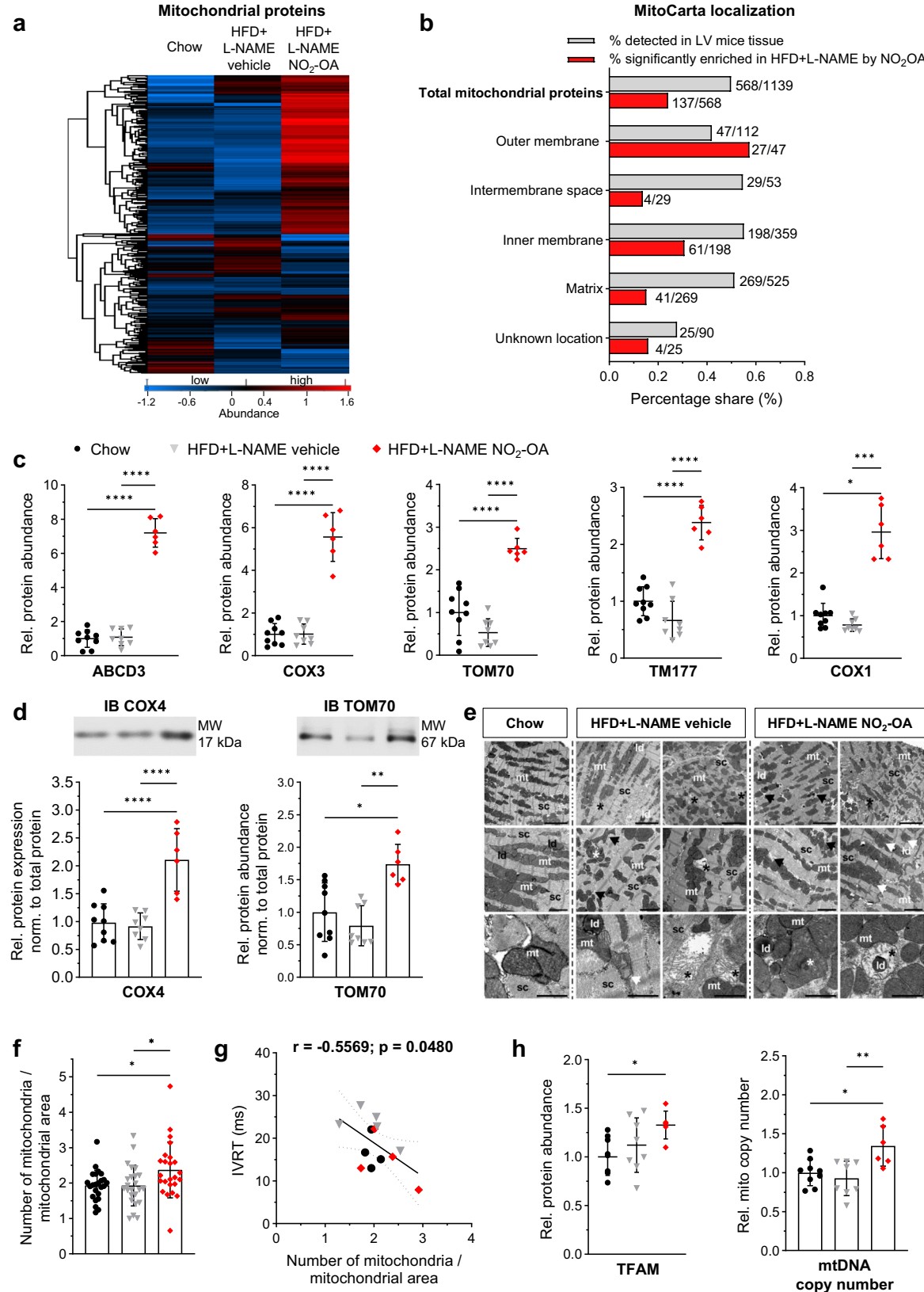

well as compensatory effects of SIRT1 and SIRT3 protein expression[38,39] have been reported. It remains to be defined what specific mechanisms account for AMPK and SIRT pathway modulation by NO₂-OA. PPARα agonism by NO₂-OA might be of relevance, given the role of PPARα in stimulating FAO in the heart; however, the partial agonism of NO₂-OA for PPARγ appears greater than that for PPARα[40]. Systemic PPARγ

activation is likely to account for some anti-hyperglycemic effects in NO₂-OA-treated mice, as previously[24]. However, the improvement in glucose tolerance is unlikely to be responsible for reducing diastolic dysfunction. Indeed, clinical data shows that the normalization of blood glucose in diabetic patients does not reduce cardiovascular death[41], and thiazolidinediones (full PPARγ agonists) increase the risk

**Fig. 3 | Mitochondrial dynamics is stimulated by nitro-oleic acid in LV of mice with HFpEF.** Identification of mitochondrial proteins (**a**) and localization (**b**) by mitoCarta[29]. **a** Heatmap of mitochondrial proteins detected by liquid chromatography–mass spectrometry in left ventricular tissue of mice after chow diet, 15 weeks of high-fat diet (HFD) and L-NAME with 4 weeks of vehicle (HFD + L-NAME vehicle) or $NO_2$-OA treatment (HFD + L-NAME $NO_2$-OA). Protein names and z-scores are in the source data file. **b** Shown is the percentage of identified mitochondrial proteins (grey) and the percentage of mitochondrial proteins with significantly enhanced abundance after administration of $NO_2$-OA in HFD + L-NAME mice compared to vehicle-treated HFD + L-NAME mice (red). **c** Protein levels of highly abundant mitochondrial proteins ($-\log_{10}(p\text{-value}) > 4.5$; $\log_2(\text{fold change}) > 1.9$) ($N = 9/8/6$), ABCD3: ATP-binding cassette subfamily D member 3; COX: cytochrome c oxidase; TOM70: translocase of outer mitochondrial membrane 70; TM177: transmembrane protein 177. **d** Representative immunoblots (IB) and quantification of COX4 and TOM70 ($N = 9/8/6$). Protein levels were normalized to

the total protein amount assessed on IB by fluorescence labeling (Figs. S12 and S13). **e** Representative TEM images ($N = 5$ animals per group) at magnifications of 5 K (scale bar 5 μm), 10 K (scale bar 2 μm), and 31.5 K (scale bar 1 μm), sc: sarcomere; ld: lipid droplet; mt: mitochondria with intact cristae; white arrow: early cristae fragmentation; white asterisk: cristae adhesion; black asterisk: swollen, ruptured mitochondria. **f** Number of mitochondria normalized by mitochondrial area (24 images per group, $N = 4/5/4$). **g** Correlation of **f** with isovolumetric relaxation time (IVRT). **h** Relative mitochondrial DNA (mtDNA) copy number and protein abundance of the mitochondrial transcription factor (TFAM). Data are mean ± standard deviation for **c**, **d**, **f** and **h**. Data in **c**, **d** and **h** are shown relative to the mean of the chow group. Statistical significance was calculated with One-way ANOVA followed by Bonferroni´s post-hoc test for **c**, **d**, **f** and **h**. Kruskal-Wallis-test followed by Dunn´s multiple comparison test was used for COX1 in **c** and TOM70 in **d**. Pearson correlation was used for **g**. Only statistically significant differences are indicated. $p$: *<0.05, **<0.01, ***<0.001, ****<0.0001.

of heart failure despite effective blood glucose reduction in diabetic patients[42,43].

The present data reveals the relevant actions of $NO_2$-OA in obese hypertensive mice in the context of profoundly enhancing mitochondrial dynamics. The increased mitochondrial protein mass, in particular outer membrane proteins, together with more but smaller mitochondria, suggests that mitochondrial biogenesis, fission, and mitophagy might be upregulated by $NO_2$-OA, all of which are well-known consequences of AMPK activation[35]. TEM analyses revealed impaired mitochondrial integrity in both HFD + L-NAME-treated groups. However, a shift towards smaller mitochondria is described as a compensatory mechanism to enhance mitochondrial productivity counteracting damaging processes and has been related to the preservation of myocardial energetics and diastolic function[44]. Thus, $NO_2$-OA-dependent AMPK activation may instigate the removal of damaged mitochondria and the stabilization of intact organelles. The observation of reduced glycogen in $NO_2$-OA- compared to vehicle-treated mice in TEM images further supports the activation of AMPK, which prevents glycogen storage[45]. In murine cardiomyocytes exposed to hyperglycemic metabolic stress during electrical pacing with AMPK activation, increased TOM70 protein and OXPHOS levels were evident. Small changes in some parameters might be related to the short-term treatment of 48 h due to the challenges of the long-term culture of isolated adult cardiomyocytes. Nonetheless, the results reveal a direct effect of $NO_2$-OA on cardiomyocytes under conditions established for modeling diabetic cardiomyopathy[33].

Given that in metabolic syndrome, FAO is increased due to lipid oversupply and resulting insulin resistance, it has been hypothesized that a reduction of FAO is beneficial in cardiometabolic HFpEF. However, this hypothesis has not been supported by preclinical and clinical studies. In HFD-fed mice with inducible deletion of acetyl coenzyme A carboxylase 2, enhanced FAO protected from the development of HFpEF[46]. In this regard, the DoPING-HFpEF trial showed that inhibiting FAO with trimetazidine in HFpEF patients did not improve post-capillary pulmonary hypertension or exercise capacity[47]. Instead, metabolic flexibility, rather than FAO suppression, appears to be critical for maintaining cardiac function. This is partly because FA are efficient energy substrates, coupling energy production to the high metabolic demands of HFpEF. Energetic deficits are a hallmark of HFpEF, characterized by reduced PCr/ATP ratios or ATP/ADP ratios[48]. In HF models using isolated cardiomyocytes or whole hearts, these deficits are associated with increased LVEDP, impaired myocardial relaxation, elevated cellular contractility, and increased calcium transients[49–52]. These findings align with a recent study in a comparable HFpEF mouse model (HFD + L-NAME), where cardiomyocytes exhibited increased contractility alongside elevated calcium transients[53]. Reduced ATP/ADP ratios in cardiomyocytes exacerbate sarcomeric dysfunction by promoting the formation of force-producing cross-bridges, delaying calcium reuptake, and diminishing ATP hydrolysis

efficiency[49–52]. Interestingly, creatine kinase activity, a key regulator of cardiac energy homeostasis, has been shown to correlate positively with AMPK activation under normoxic conditions, linking improved energy metabolism to better cardiac performance[54]. $NO_2$-OA's ability to activate AMPK and enhance mitochondrial metabolism aligns closely with these findings, emphasizing its therapeutic potential in addressing HFpEF-related energetic deficits. In line, SGLT2 inhibitors have been proposed to exert physiological benefit by inducing ketone metabolism[55], and AMPK activation by metformin exerts cardioprotective effects in HFpEF[56]. Furthermore, given that AMPK induces mobilization of lipid stores, FAO, and mitochondrial dynamics, not only PCr/ATP ratio in cardiomyocytes is increased, but also the turnover of high levels of FAs are enhanced and thus lipotoxicity is reduced. Herein, LC-MS analysis of lipids did not detect a reduction of DAG, triacylglycerols (TAG), and ceramides in myocardial tissue of $NO_2$-OA-treated mice, but in isolated cardiomyocytes of HFpEF mice the amount of lipid droplets was reduced by $NO_2$-OA compared to vehicle. Thus, lipid droplets, which reflect the cardiomyocyte lipid storage, appear to be reduced consistent with AMPK-mediated lipid catabolism, whereas lipidomics from myocardial homogenates detect lipid levels also from other cells than cardiomyocytes and other compartments than lipid droplets. The intensified lipid turnover and mitochondrial dynamics can promote the improvement of myocardial bioenergetics, thus restoring cardiomyocyte relaxation and mitigating heart failure symptoms, as reflected by improved exercise capacity.

The fact that the work was done on two completely independent cohorts of mice sourced from two different vendors, with the second cohort confirming and validating the phenotypic data, significantly strengthens the results. Importantly, allometric scaling reveals that the $NO_2$-OA dose used in this study is at the lower end of what the Phase II clinical trial tested. Higher doses, which are safe in humans, might potentially exert even stronger effects. Together with the already recognized beneficial actions of $NO_2$-OA on cardiac and metabolic disorders, small-molecule nitroalkenes may present a unique repertoire of actions that could pharmacologically impact clinical HFpEF therapy.

This preclinical study has limitations. Most importantly, we have not determined PCr/ATP levels specifically in the LV of mice. Second, we suggest that lipid turnover is increased by $NO_2$-OA, which we have not measured directly. Therefore, defining changes in lipid handling and its effect on myocardial energetics is a future goal. Furthermore, we have not analyzed calcium handling, which is important to fully understand the diastolic dysfunctional phenotype. Another important point is that we have not measured blood pressure in conscious mice. It has been previously reported that $NO_2$-OA inhibits angiotensin II (AngII) type 1 receptor (AT1R) signaling, thereby reducing blood pressure in AngII-treated mice[57]. Furthermore, $NO_2$-OA inhibits soluble epoxide hydrolase, which leads to epoxyeicosatrienoic acid-mediated lowering of blood pressure[58]. However, systemic vascular resistance

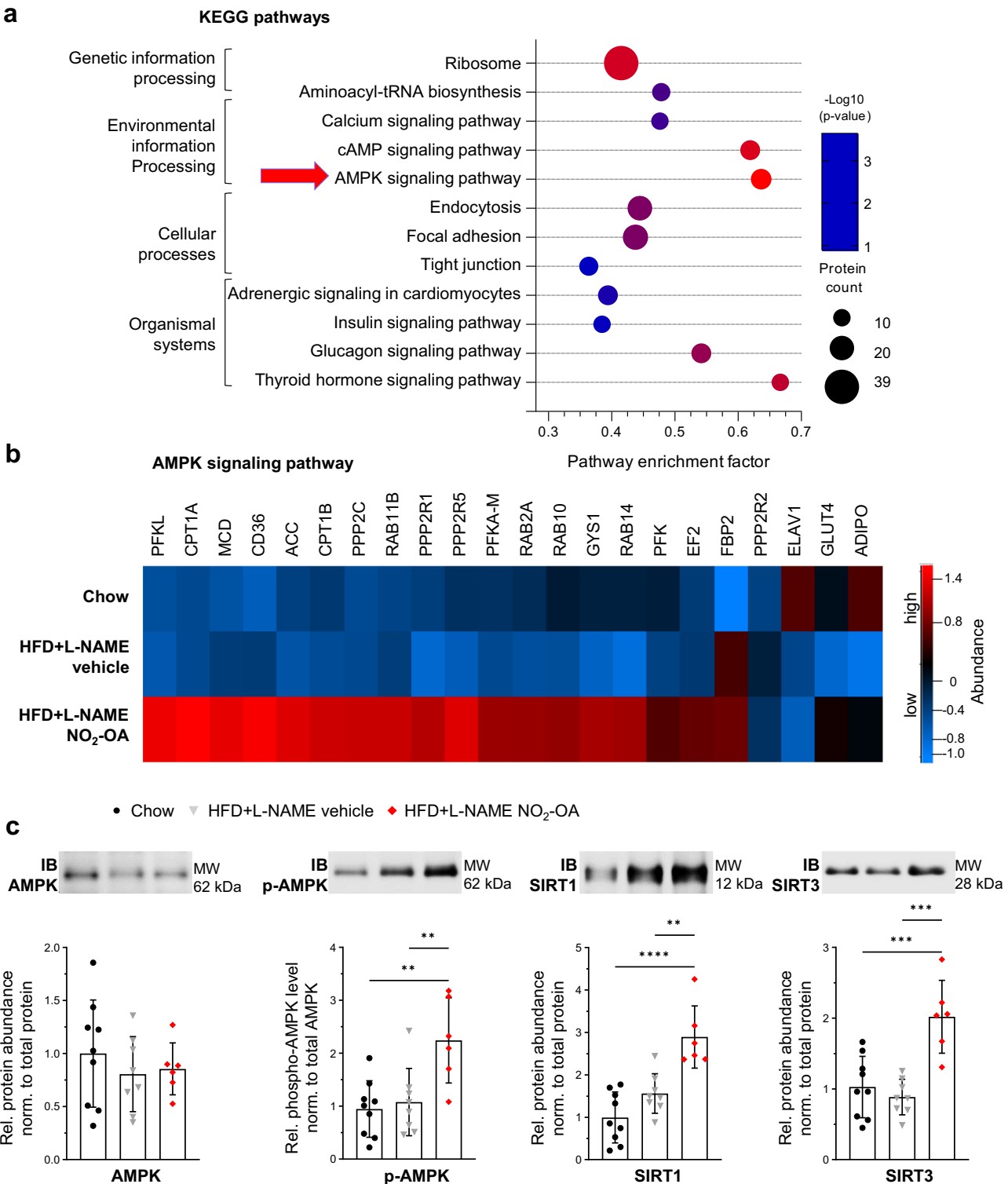

**Fig. 4 | Activation of the AMPK signaling pathway is induced by nitro-oleic acid in LV of HFpEF mice. a** KEGG pathway enrichment analysis of proteins derived from liquid chromatography followed by mass spectrometry (LC-MS) with significantly enhanced abundance after administration of $NO_2$-OA in high-fat diet (HFD) + L-NAME treated mouse hearts compared to vehicle-treated mouse hearts (HFD + L-NAME vehicle) (inclusion criteria: protein count ≥ 10; pathway enrichment factor ≥ 30%; significance ≥ -$\log_{10}$(0.05)). The most significant KEGG pathway is marked with a red arrow. **b** Heatmap of all proteins associated with the AMP-activated protein kinase (AMPK) signaling pathway and identified by LC-MS

analysis. Protein names and z-scores are in the source data file. **c** Representative immunoblots (IB) and quantification of AMPK and phosphorylation level of AMPK at threonine residue 172 (p-AMPK), sirtuin 1 (SIRT1) and sirtuin 3 (SIRT3) ($N$ = 9/8/6). Protein levels were normalized to the total protein amount assessed on IB by fluorescence labeling (Figs. S12 and S14). Data in **c** are shown relative to the mean of the chow group as mean ± standard deviation. Statistical significance was calculated by two-tailed Fisher's exact test for **a** and One-way ANOVA followed by Bonferroni´s post-hoc test for **c**. Only statistically significant differences are indicated. $p$: **<0.01, ***<0.001, ****<0.0001.

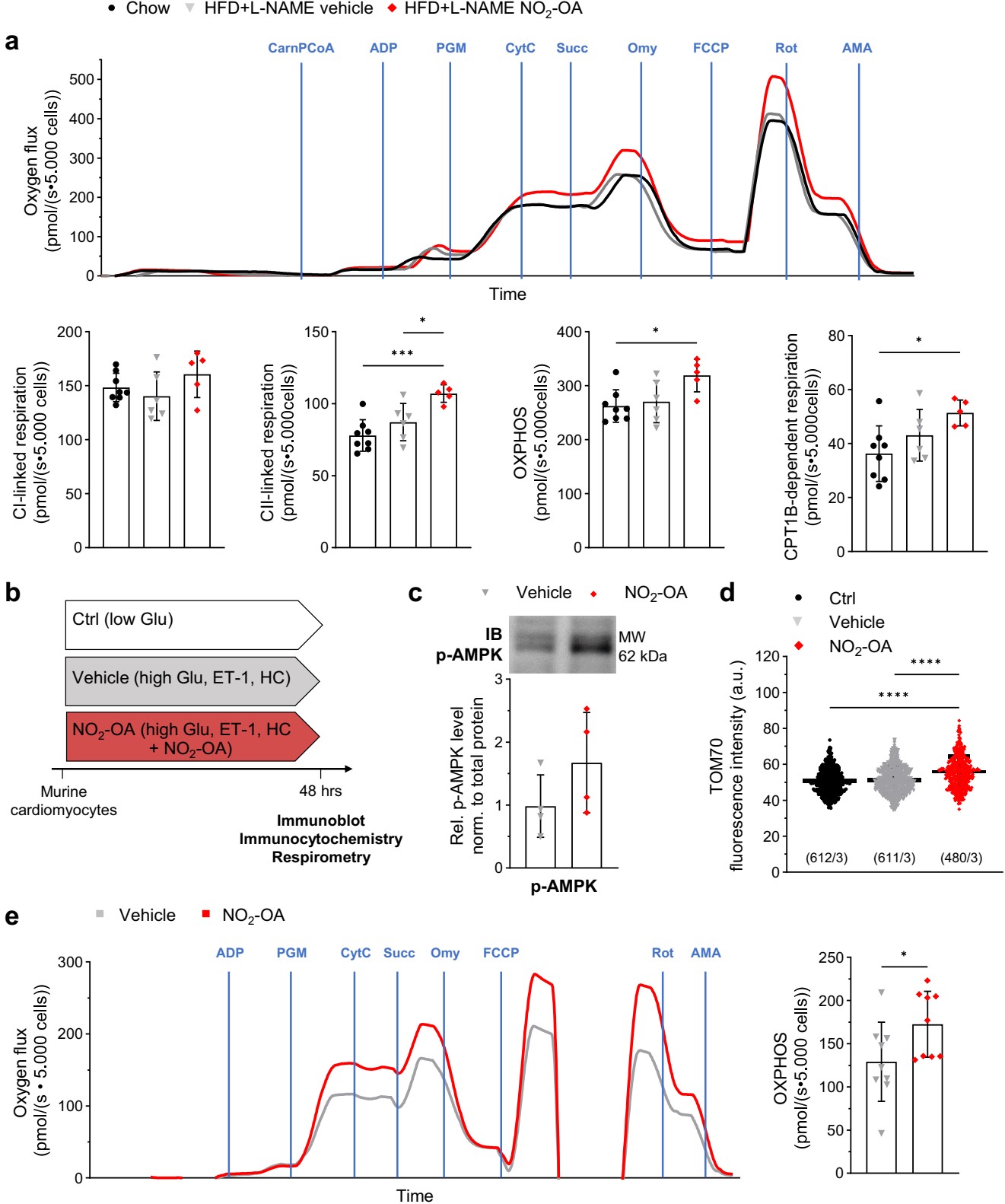

and end-systolic LV pressure were not decreased by NO$_2$-OA in hemodynamic analyses. Apart from that, the multi-target signaling actions of NO$_2$-OA can exert protection from diastolic dysfunction in this model, i.e., enhanced activation of the ryanodine receptor[59], modulation of mitochondrial function by nitroalkylation of uncoupling proteins (UCP) and adenine nucleotide translocase (ANT)[60,61], reduction of adipocyte dysfunction via anti-inflammatory mechanisms and other potential actions.

Murine modeling of HFpEF also presents limitations. Herein, the impact of age was not considered as we used young mice. Not only inflammation and vascular function, but also mitochondrial metabolism and inter-organ communication is different in aged individuals. Furthermore, 15 weeks is a short period of the murine life span compared to HFpEF development in patients. Thus, a longer time period may engender more advanced fibrotic and inflammatory remodeling, endothelial dysfunction and metabolic dysbalance, thus

**Fig. 5 | Nitro-oleic acid enhances cardiac mitochondrial respiration.** Cardiomyocytes of mice after chow diet, 15 weeks (wk) of high-fat diet (HFD) and L-NAME with 4 wk of vehicle (HFD + L-NAME vehicle) or NO2-OA treatment (HFD + L-NAME NO2-OA) were isolated and subjected to high resolution respirometry.
**a** Representative trace of oxygen flux (upper panel). Addition of components are marked with blue lines, CarnPCoA: carnitine palmitoyl-CoA; ADP: adenosine diphosphate; PGM: pyruvate glutamate malate; CytC: cytochrome C; Succ: succinate; Omy: oligomycin; FCCP: mitochondrial uncoupler; Rot: rotenone; AMA: antimycin A. Quantification of complex I- and complex II-linked respiration, oxidative phosphorylation (OXPHOS), and carnitine palmitoyltransferase 1 (CPT1B)-linked respiration (lower panel) ($N = 8/6/5$). **b** Isolated adult murine cardiomyocytes were cultivated for 48 h under control conditions (Ctrl) with normal glucose levels (low Glu: low glucose levels), under metabolic stress conditions induced by high glucose levels (high Glu), endothelin-1 (ET1) and hydrocortisone (HC) with treatment of methanol (Vehicle) or nitro-oleic acid (NO2-OA). **c** Representative

immunoblot (IB) and phosphorylation level of AMP-activated protein kinase at threonine residue 172 (p-AMPK). Protein levels were normalized to the total protein amount assessed on IB by fluorescence labeling (Fig. S15) and shown relative to the individual Ctrl group ($N = 4$). **d** Quantification of fluorescence intensity of translocase of outer mitochondrial membrane 70 (TOM70), shown as arbitrary units (a.u.), for all analyzed individual cardiomyocytes of three experiments ($N = 3$). **e** High resolution respirometry was performed with primary murine cardiomyocytes after cultivation and treatment. Representative trace of oxygen flux. The addition of components are marked with blue lines. Oxidative phosphorylation (OXPHOS) values were extracted from oxygen flux traces ($N = 9$). Data are presented as mean ± standard deviation. Statistical significance was calculated by One-way ANOVA followed by Bonferroni´s post-hoc test for **a** and **d** and unpaired, two-sided Student's t-test for **e**. Only statistically significant differences are indicated. $p$: *<0.05, ***<0.001, ****<0.0001.

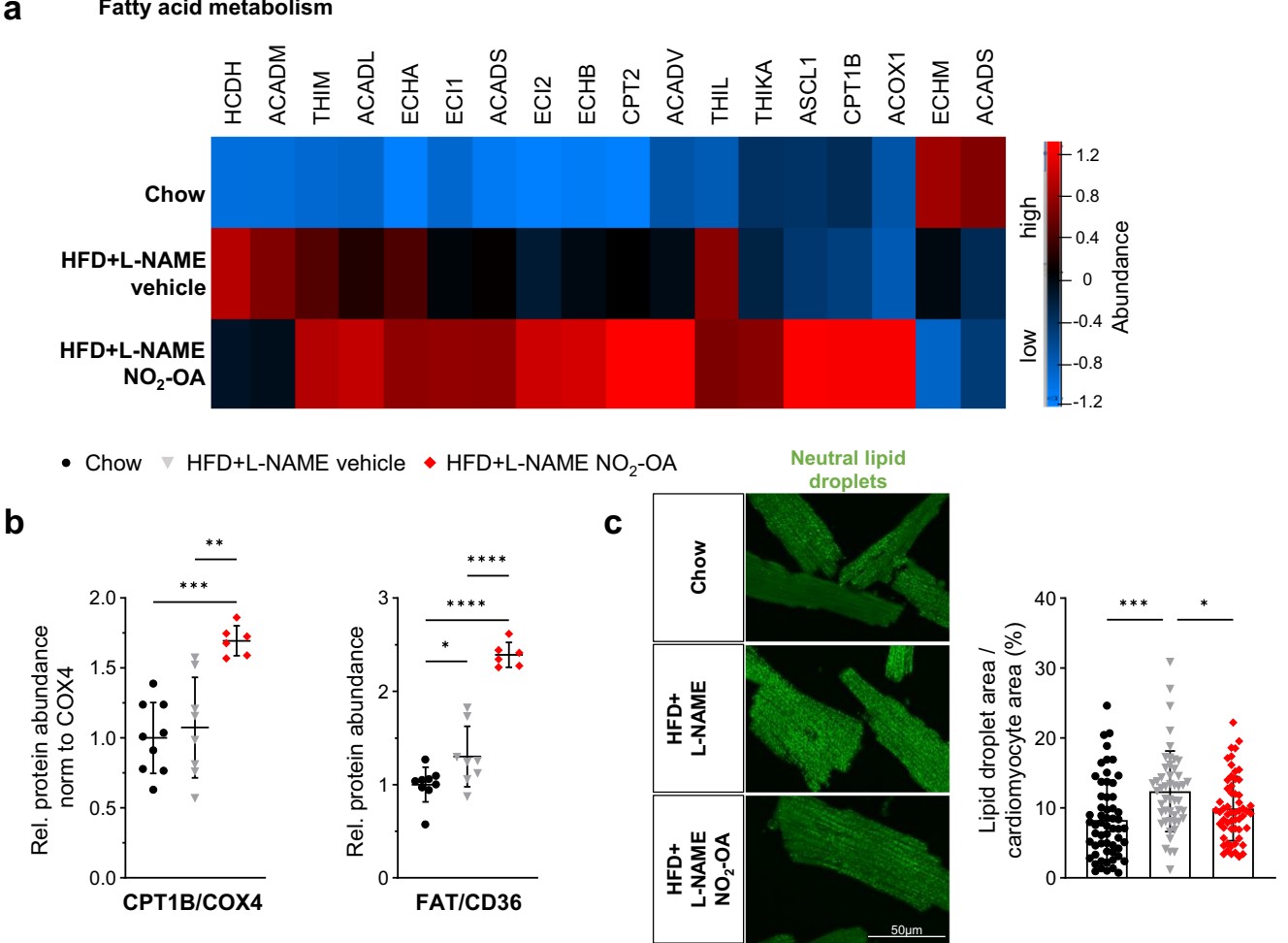

**Fig. 6 | Nitro-oleic acid enhances cardiac fatty acid metabolism in HFpEF mice.**
**a** Heatmap of all proteins associated with fatty acid metabolism and identified by liquid chromatography mass spectrometry (LC-MS) after administration of NO2-OA in high-fat diet (HFD) + L-NAME treated mouse hearts compared to vehicle-treated mouse hearts. Protein names and z-scores are in the source data file. **b** Protein abundances of carnitine palmitoyltransferase 1B (CPT1B) relative to cytochrome c oxidase 4 (COX4) and long chain fatty acid transporter (FAT/CD36) derived from LC-MS. Data are shown relative to the mean of the chow group ($N = 9/8/6$).

**c** Representative images of adult murine cardiomyocytes stained for neutral lipids (left). Quantification of lipid droplet area normalized to cardiomyocyte area shown for all individual analyzed images (right) (9 to 10 images at a magnification of 63x per animal, $N = 6/5/6$). Data are presented as mean ± standard deviation. Statistical significance was calculated by One-way ANOVA followed by Bonferroni´s post-hoc test. Only statistically significant differences are indicated. $p$: *<0.05, **<0.01, ***<0.001, ****<0.0001.

presenting additional opportunities for defining drug reversal of cardiac pathology. Mice are housed under germ-poor conditions, which might have an impact on the immune system and the microbiome, both of which play an important role in heart disease pathophysiology.

Finally, the proteomics approach has a few methodological limitations. First, high molecular-weight proteins might be underrepresented among all proteins that were isolated from the myocardial tissue, as displayed by staining total proteins in a molecular weight range after SDS PAGE (Fig. S16). Second, the selection of

proteins used for data interpretation was very strict omitting proteins from the analysis if they were not detectable in every sample. However, key results of the proteome analysis were successfully validated by other methods.

Our study has identified enhanced mitochondrial biogenesis and metabolism to be critical for restoring myocardial relaxation and reducing heart failure symptoms in a murine model of obese HFpEF. At clinically relevant doses being evaluated in a Phase II clinical trial, the small molecule nitroalkene NO$_2$-OA induced these effects, which were associated with activation of AMPK signaling. Given the limited treatment options for HFpEF, these findings underline the promising biological properties of NO$_2$-OA. An ongoing phase 2 trial testing NO$_2$-OA in obese asthmatics (NCT03762395) is revealing patient safety and insight into the gene expression, metabolomic, inflammatory signaling, and physiological responses that small molecule nitroalkene induces in obese subjects. Despite the promising effects of SGLT2 inhibitors in HFpEF patients[5] and the impressive effects of GLP-1 agonists on symptom burden in obese HFpEF[62], more treatment options are warranted for this highly prevalent and morbid disorder, especially considering the limited tolerability to GLP-1 agonists and their impact on muscle wasting[63]. With a favorable safety and tolerability profile, NO$_2$-OA emerges as a potentially important drug for treating HFpEF and other cardiopulmonary diseases[64].

## Methods

### Animals
Male C57BL/6 N mice were purchased from Charles River (Sulzfeld, Germany) for an initial cohort and from Janvier Labs (Le Genest-Saint-Isle, France) for a second cohort. Female mice were from in-house breeding based on C57BL/6 N (Janvier Labs). Male mice were housed for at least 1 week to acclimatize to laboratory conditions before starting any experimental procedure. All mice were kept in a 12:12 h inverse light cycle in ventilated cabinets that were maintained at $23 \pm 2\,°C$ and 40–60% humidity. Routine tests were performed to ensure that mice are pathogen-free according to the guidelines of the Federation of European Laboratory Animal Science Associations (FELASA). Mice had unrestricted access to water and food. For organ extraction, mice were analgized with buprenorphine (0.1 mg/kg BW) and anesthetized using isoflurane inhalation.

All animal studies were approved by the local Animal Care and Use Committees (Ministry for Environment, Agriculture, Conservation and Consumer Protection of the State of North Rhine-Westphalia: State Agency for Nature, Environment and Consumer Protection (LANUV), NRW, Germany) and followed guidelines from Directive 2010/63/EU of the European Parliament on the protection of animals used for scientific purposes.

### Animal protocol- male mice
Cardiometabolic syndrome was induced in male mice by HFD combined with the established eNOS inhibitor L-NAME, as described by Schiattarella et al.[25] 4-week-old male mice were fed an HFD (60% fat, 20% protein, 20% carbohydrates; E15742-34; Ssniff, Germany) or standard rodent diet (chow group; V1534-703; Ssniff, Germany). L-NAME (0.5 g/L; #N5751; Sigma Aldrich) was supplied in the drinking water. The HFD + L-NAME and chow groups were fed for 11 weeks and body weight was determined every week.

After 11 weeks, the HFD + L-NAME group was divided either into 3 groups for some experiments (vehicle, NO$_2$-OA, or oleic acid (OA); Fig. S1; male mice Charles River, initial cohort) or into 2 groups (vehicle or NO$_2$-OA; Fig. 1, male mice Janvier Labs, second cohort). Nitro-oleic acid (50:50 mix of (E)-9- and 10-nitrooctadec-9-enoic acid; NO$_2$-OA, was synthesized in-house[40], oleic acid (OA) or vehicle (polyethylene glycol:ethanol 90:10) were administered with ALZET mini-osmotic pumps (NO$_2$-OA and OA 8.17 mg/kg bw/day, Model 2002, ALZET, Cupertino, CA, USA) for 4 weeks. After final investigations at 15 weeks, organs were harvested and snap-frozen for RNA and protein analysis or fixed in formalin or processed as described below for further analysis. The NO$_2$-OA selected dose correspond to a 46 mg daily dose in humans by allometric scaling. The Phase II clinical trials in humans used daily doses that range from 75 to 300 mg of a specific NO$_2$-OA isomers per day.

### Animal protocol - female mice
Cardiometabolic syndrome was induced in female mice by HFD combined with the established eNOS inhibitor L-NAME. 8-week-old female mice were fed a HFD (60% fat, 20% protein, 20% carbohydrates; E15742-34; ssniff, Germany) or standard rodent diet (chow group; V1534-703; ssniff, Germany). L-NAME (0.5 g/L; #N5751; Sigma Aldrich) was supplied in the drinking water. The HFD + L-NAME and chow groups were fed for 15 weeks and body weight was determined every week.

After 11 weeks, all female HFD + L-NAME mice were treated with peanut butter pellets with or without nitro-oleic acid (50:50 mix of (E)-9- and 10-nitrooctadec-9-enoic acid; NO$_2$-OA, was synthesized in-house[40], 15 mg/kg bw/day) for 4 weeks between week 12 to15. The pellets were formulated weekly[65]. In brief, peanut butter (MisterChoc, smooth) was heated to 55 °C, and diluted nitro-oleic acid (NO$_2$-OA:ethanol 75:25) was mixed in. Pellets were formed with a pellet mold (Corticosterone Pellet Mold, Prod No. 106A, Ted Pella, Inc., Redding, USA), cooled to room temperature and stored at −80 °C. Echocardiography was performed before (Baseline) and after treatment with NO$_2$-OA (Final).

### Echocardiography
Cardiac ultrasound was carried out with a Vevo 3100 Imaging System (FUJIFILM Visualsonics, Inc., Toronto, ON, Canada) using a MX550D transducer (25–55 MHz, FUJIFILM Visualsonics, Inc., Toronto, ON, Canada). Mice were anesthetized with isoflurane inhalation (1.5-2%) and placed in supine position on a heating pad. An electrocardiogram was obtained with integrated electrodes. Body temperature was monitored using a rectal probe (T = 36.5–37.5 °C). Respiration rate (RR = 80–120) was controlled by adjusting the depth of anesthesia. A standardized workflow was followed as stated in the position paper of the Working Group on Myocardial Function of the European Society of Cardiology[66]. In brief, 2D recordings of brightness- (B-) and motion-(M-) mode of parasternal long-axis (PSLAX) and parasternal short-axis (PSAX), respectively, were acquired for analysis of LV ejection fraction (LVEF) and posterior wall thickness in diastole (LVPW, d). Diastolic function was obtained in apical four-chamber view (4APIX) with pulsed-wave and tissue Doppler imaging at the level of the mitral valve. Peak Doppler blood inflow velocity across the mitral valve during early diastole (E wave) and during late diastole (A wave) and peak tissue Doppler of myocardial relaxation velocity at the mitral valve annulus during early diastole (e' wave) were assessed. All parameters were determined three times. The average is presented for each animal.

### Strain analysis
The strain analysis was carried out using the Vevo strain analysis module of the Vevo Lab analysis software (Version 3.2.0; FUJIFILM Visualsonics, Inc., Toronto, ON, Canada). Strain determines the deformation of the myocardial wall and thereby reflects cardiac function. The analysis uses two-dimensional speckle tracking to calculate the distance a speckle moves between two consecutive frames, which is indicated as displacement. The velocity is the displacement per unit time in three planes. The longitudinal strain is a dimensionless parameter and shows the tangential movement based on the traced border normalized towards a baseline. The software uses the Lagrangian Strain algorithm, which is based on the following formula: $S(t) = (L(t) − L(0)) / L(0)$; $S(t)$, Strain value at a specific time point $t$; $L(t)$, length at a specific time point $t$; $L(0)$, length at the start point 0. The

analysis was performed by using three cardiac cycles of recordings of parasternal long-axis (PSLAX) view.

## PV-Loops

Mice anesthetized with isoflurane (Baxter, Deerfield, IL, USA) received low-dose buprenorphine (Sigma-Aldrich, St. Louis, MO, USA; 0.05 mg/kg bodyweight subcutaneously) for analgesia and were placed on a heating pad to maintain body temperature. Following endotracheal intubation, animals were ventilated with a peak inspiratory pressure (PiP) of 18 mmHg and a positive end-expiratory pressure (PEEP) of 2 mmHg with 140 strokes/min using a VentElite mouse ventilator (Harvard Apparatus, Holliston, MA, USA). Tidal volume was constantly adjusted automatically. The left jugular vein was cannulated with PE-10 tubing and a solution of 12.5% bovine serum albumin (Sigma-Aldrich, St. Louis, MO, USA) was infused. Heart rate was maintained between 400 and 500 bpm by adjusting the concentration of isoflurane accordingly. The thorax was opened, the apex was punctured with a 27 G cannula, and a 1.0 F microtip conductance catheter was inserted into the left ventricle. Left ventricular (LV) pressure and volume were recorded with an ADInstruments PowerLab C system (ADInstruments, Dunedin, New Zealand). Saline (Carl Roth, Karlsruhe, Germany, 10%) was injected to correct volume for parallel conductance. Volume calibration was performed afterwards using ADInstruments volume calibration cuvette. Analysis was carried out in Circlab (v12.5, Prof. P. Steendijk, Leiden University). Chamber stiffness was calculated using a single beat estimation according to Burkhoff et al.[67].

## Graded exercise test

A graded exercise test was carried out using an automated protocol on a Panlab Treadmill (Model: LE8708TS Airtight, Panlab, Harvard Apparatus, Holliston, MA, USA). The mice were acclimatized to the treadmill in three consecutive training sessions at least 2 days apart. One week after the last acclimatization session, the exercise exhaustion test was performed. Mice ran uphill with an incline of 20° starting with a speed of 8 cm/s for 4 min after which speed was increased to 23 cm/s for 2 min. Every subsequent 2 min, the speed was further increased by 3 cm/s until the mouse was exhausted. Exhaustion was defined as the inability of the mouse to return to running within 5 s of direct contact with an electric-stimulus grid. Running time and distance were measured. Work rate was calculated for every stage of the exercise protocol for every mouse according to the following formula: $WR(t) = BW \cdot v \cdot \sin(\alpha) \cdot g$; $WR(t)$, work rate; BW, body weight in kg; v, speed; $\alpha$, incline in radians; g, gravital force. The sum of the work rate per stage of the exercise protocol is presented.

## Glucose-tolerance test

Glucose-tolerance tests were performed before final harvest at 15 weeks of HFD + L-NAME. After 6 h fasting, a glucose solution (2 g/kg in saline) was injected intraperitoneally. Tail blood glucose levels (mg/dL) were measured with a glucometer (B Braun Melsungen, Melsungen, Germany) before (0 min) and at 15 min and 90 min after glucose administration. The area under the curve (AUC) was calculated by using the series of determined measurements over time.

## Liquid chromatography-mass spectrometry (LC-MS)

The LV tissue was ground in liquid nitrogen to a fine powder. After incubation for 10 min in 4% SDS Tris-Cl buffer (pH 7.6) and centrifugation (16,000 g for 15 min), the supernatants were diluted with 4 volumes of ice-cold acetone to precipitate the proteins (overnight at −20 °C). The proteins were redissolved in 8 M Urea/50 mM TEAB buffer containing protease- and phosphatase inhibitors (EDTA free cOmplete™, PhosSTOP™ Roche) and treated 30 min at 37 °C with 50 units/100 μg tissue benzonase (Sigma Aldrich) to degrade chromatin. A BCA Assay (Pierce) was used to determine the protein concentration. After reduction of each 100 μg Protein (5 mM DTT for 1 h at 25 °C) and alkylation

(40 mM iodacetamide for 30 min at 25 °C in the dark), Lys-C (Promega) was added in an enzyme:substrate ratio of 1:75 for 4 h at 25 °C. The samples were diluted with 50 mM TEAB buffer to a final Urea concentration of 2 M before adding trypsin (Promega) in an enzyme:substrate ratio of 1:75 and incubating at 25 °C for 16 h. The enzymatic digestion was stopped by adding formic acid to a final concentration of 1%. The stop and Go extraction (Stage) was used to prepare the protein digestions for mass spectrometric analysis[68]. Two small pieces of Empore™ SDB-RPS extraction disks (Supelco, 66886-U) were placed with a 16 gauge blunt-ended syringe needle in 200 μl pipette tips. StageTips were conditioned by 60 μl of methanol and centrifugation at 450 g for 75 s, followed by centrifugation with 60 μl buffer B (0.1% formic acid / 80% acetonitrile) and twice with 60 μl buffer A (0.1% formic acid) for equilibration. The samples were loaded and passed through the StageTips by centrifugation at 450 g for 75 s, followed by washing steps with 60 μl buffer A and two times with 60 μl buffer B. After air drying, the samples were eluted two times with 60 μl 60:35:5% acetonitrile:water:ammonium hydroxide, vacuum dried, and redissolved in 50% acetonitrile / 0.1% trifluoroacetic acid in a concentration of 1 μg peptide/μl.

The peptides were analyzed using a nanoLC (Ultimate 3000, Thermo Fisher Scientific, Germany) coupled to a modified ESI-Orbitrap MS/MS (QExactive Plus, Thermo Scientific™, Walthem, MA, USA) equipped with a Spectroglyph source (Spectroglyph, LLC, WA, US) including an ion funnel instead of an S-lense[69]. 1 μg of protein from each sample was loaded onto the enrichment column before switching in line with the analytical column at a flow rate of 300 μL/min. The gradient length of the Acclaim™ PepMap™ C18 2 μm 75 μm × 500 mm (Thermo Fisher Scientific, Dreieich, Germany) analytical column was adjusted to 187 min from 4 to 50% of 80% acetonitrile and 0.08% formic acid at a flow rate of 300 nl/min. ESI-Orbitrap mass spectrometry measurements were carried out in a data-dependent top-10 acquisition mode. All samples were measured in full MS mode using a resolution of 70,000 (AGC target of 3e6 and 64 ms maximum IT). For the dd-MS2, a resolution of 17,500, AGC target of 2e5 and a maximum IT of 200 ms were used.

The Proteome Discoverer 2.5 software (Thermo Fisher, Dreieich, Germany) was used for database search against the Mus musculus proteome database (UniProt, v2022-03-02) and the PD_Contaminants database (v2015_5). The enzyme specificity was set to trypsin with a maximum number of two missed cleavages with a fragment tolerance of 0.02 da and a precursor mass tolerance of 10 ppm. For the variable modifications, methionine oxidation and N-terminal acetylation were set. Fixed modifications were set for the carbamidomethylation of cysteine. Match between runs was enabled. Only unique peptides were included in the analysis. The minimum false discovery rate (FDR) at which a peptide-spectrum match (PSM) was considered significant was set to 0.01. No technical replicates were measured. Proteins considered as potential contaminants were deleted. In total 2294 proteins were identified with a minimum of two unique peptides (Supplementary Data 1) in left ventricular tissue of 9 control mice (Chow), 8 mice who received high-fat diet and the eNOS inhibition by L-NAME for 15 wk and were treated with vehicle for the last 4 wk (HFD + L-NAME vehicle), and 6 HFD + L-NAME mice, who were treated with NO$_2$-OA for the last 4 wk (HFD + L-NAME NO$_2$-OA).

For data visualization and statistics, the Perseus software tool version 2.0.9.0 (MPI Biochemistry) was used[70]. Only proteins with valid protein abundance in all biological samples were considered. According to these criteria, 1878 proteins were considered for post hoc analysis with Perseus and data interpretation (source data file). Differentially expressed proteins were defined as those with a fold change > 1.5 and FDR < 0.05 (Fig. 2b and d). The individual values for each protein are displayed in the source data file.

## Cardiomyocyte isolation

Murine cardiomyocytes were isolated using a modified Langendorff apparatus[71]. The perfusion buffer (113 mM NaCl, 4.7 mM KCl, 0.6 mM

$Na_2HPO_4 2H_2O$, 0.6 mM $KH_2PO_4$, 12 mM $NaHCO_3$, 10 mM $KHCO_3$, 1.2 mM $MgSO_4 7H_2O$, 10 mM HEPES, 30 mM taurine, 10 mM BDM, 10 mM glucose), digestion buffer (100 μg/mL Liberase TM (Roche), 0.0125 mM $CaCl_2$ in perfusion buffer) and stop buffers (increasing calcium and BSA concentration, final concentration 1.5 mM $CaCl_2$ and 10% BSA) were adapted to achieve a maximum yield of rod-shaped cardiomyocytes.

The aorta was canulated and the heart flushed retrograde with perfusion buffer for 3 min and then with digestion buffer for 3–10 min, depending on the drip speed. The digestion was assumed to be completed when the drip speed reached 3–5 drops per second. The heart was removed from the cannula, poured over with stop buffer, and the atria and vessels were removed and sliced into small pieces. The suspension was filtrated through a 200 μm mesh, and cells were allowed to settle by gravity. Afterwards, cells were purified with solutions of increasing calcium concentrations (0.075, 0.225, 0.6, 1.5 mM).

## Cardiomyocyte culture

The primary mouse cardiomyocytes were cultured for immunofluorescence stainings on laminin (Merck, L2020) surface-coated four well Permanox slides (Thermo Fisher Sci, 177437) or for high-resolution respirometry measurements in untreated 6 well cell culture plates in culture medium (M199 Hanks (Gibco 12350-039), 5 mM creatine, 2 mM carnitine, 5 mM taurine, SITE + 3 (Sigma S5295), 15 μM blebbistatin (Toronto Research Chemicals B592500)). To simulate a diabetic milieu[32], the glucose concentration was increased to 10 mM, and 10 nM endothelin-1 and 1 μM hydrocortisone were added together with or without 1 μM $NO_2$-OA. Methanol was used as vehicle. Culture was continued at 37 °C for 48 h. Cells for respirometry measurements underwent electric stimulation during culture in the 6 well plates (C-pace EM, IonOptix, Westwood, MA, USA) with 1 Hz (10 Volt, 5 msec).

## Immunoblotting

Proteins from LV tissue were isolated for immunoblotting as described above (LC-MS). For protein isolation from isolated cardiomyocytes, a two-step discontinuous Percoll (17089101, cytiva) gradient (high density 1.086 g/ml, low density 1.060 g/ml) was used to separate the pure fraction of intact cardiomyocytes. After 45 min of centrifugation at 1800 g, the band with intact cardiomyocytes between the two layers was collected and washed three times with 4 ml PBS. The cell pellet was lysed in 50 μl RIPA buffer (150 mM NaCl, 5 mM EDTA, Tris-Cl pH 8.0, 1% NP-40, 0.5% sodium deoxycholate, 0.1% SDS) supplemented with protease inhibitor (EDTA free cOmpleteTM, Roche) as well as phosphatase inhibitor (PhosSTOPTM, Roche) for 15 min on ice, two times shock frozen in liquid nitrogen and afterwards centrifuged at 20,000 g at 4 °C. Protein concentrations were determined using Pierce BCA Protein Assay Kit (23225; Thermo Fisher Scientific). Equal amounts of protein from each sample were separated on SDS–polyacrylamide gels and transferred to PVDF membrane. Membranes were stained with primary antibodies against: cytochrome-c-oxidase subunit 4 (1:5000, #4844, Cell Signaling, Danvers, MA, USA), sirtuin 3 (1:1000, #5490, Cell Signaling, Danvers, MA, USA), sirtuin 1 (1:5000; #9475, Cell Signaling, Danvers, MA, USA) phosphorylation at threonine 172 of AMP-activated protein kinase (1:500, #2531, Cell Signaling, Danvers, MA, USA), AMP-activated protein kinase (1:1000, #2532, Cell Signaling, Danvers, MA, USA), phosphorylation at serine 293 of pyruvate dehydrogenase (1:1000, #31866, Cell Signaling, Danvers, MA, USA), translocase of outer mitochondrial membrane 70 (1:2000, #14528-1-AP, Proteintech, Rosemont, Il, USA), pyruvate dehydrogenase kinase 4 (1:1000, ab214938, abcam, Cambridge, UK), peroxiredoxin-$SO_{2/3}$ (1:1000, CRB2005004, Biosynth, Staad, Switzerland), phosphorylation of serine 239 of vasodilator-stimulated phosphoprotein (1:1000, #3114, Cell Signaling, Danvers, MA, USA) and vasodilator-stimulated phosphoprotein (1:2000, #3112, Cell Signaling, Danvers, MA, USA). HRP-conjugated secondary antibodies (NA9340, VWR International GmbH, Darmstadt, Germany) and chemiluminescent substrate (WesternBright Chemilumineszenz Substrat Quantum,

Biozym, Oldendorf, Germany) were used for detection. Images were acquired using INTAS ECL CHEMOSTAR (INTAS, Göttingen, Germany), and densitometry analysis was evaluated using LabImage1D (Kapelan Bio-Imaging GmbH, Leipzig, Germany) with rolling ball background reduction. Values were normalized to total protein assessed by fluorescence staining (SPL Red, PR926, NH DyeAGNOSTICS, Halle (Saale, Germany). Data are shown relative to the mean of the control group.

## Immunohistochemistry

Pulmonary arterial muscularization was analyzed by staining of α-smooth muscle actin (A2547, 1:900, Sigma-Aldrich; 1:500 Cy3-conjugated secondary ab, 115-165-068, Jackson Immuno Research) and von Willebrand factor (PA-16634, 1:100, Invitrogen; 1:500 Alexa Fluor 488-conjugated secondary ab, A-11034, Invitrogen) on 5 μm murine lung slices. Nuclei were stained with DAPI (1:1000, ab228549, Abcam). Images were acquired with a BZ-X 810 microscope (Keyence, Osaka, Japan) and analyzed using the freeware software VessEval (https://github.com/pLeminoq/vesseval, Tamino Huxohl). αSMA-positive area of arterioles was quantified and related to vessel circumference. Arterioles with circumferences between 80 and 250 μm were included.

## Histology

Formalin-fixed, paraffin-embedded LV cross-sections of 5 μm were stained with picrosirius red (Polyscience Inc, Warrington, PA, USA) following standard protocols. Images were acquired with a BZ-X 810 microscope (Keyence, Osaka, Japan). The extent of interstitial fibrosis was graded by a blinded person in at least 6 sections of two different regions of the LV and defined as none, mild, moderate, or pronounced fibrosis.

## Insulin analysis

Islets were isolated by collagenase digestion. After lysis in acidic ethanol, insulin was determined in batches of 5 islets per sample by radioimmunoassay with rat insulin as standard.

## Immunohistochemistry for islet histology

For islet histology, paraffin-embedded 5 μm slices were stained for insulin (ab195956, 1:1500, Alexa Fluor 555-conjugated secondary antibody: ab150186, 1:500, Abcam) and glucagon (ab92517, 1:2000, Alexa Fluor 488-conjugated secondary antibody ab150077, 1:1000, Abcam). Nuclei were visualized by DAPI (Fluoroshield, Sigma Aldrich). Confocal images (10 sections, 0.35 μm distance each) were acquired and evaluated by Fiji (ImageJ).

## Lipid droplet staining

For neutral lipid staining, the isolated cardiomyocytes were incubated with 2 μM Bodipy 493/503 (4,4-Difluoro-1,3,5,7,8-Pentamethyl-4-Bora-3a,4a-Diaza-s-Indacen, D3922, Invitrogen) for 15 min at 37 °C after fixation with 4% paraformaldehyde. Images were acquired with the TCS SP8 confocal system (Leica Microsystems, Wetzlar, Germany) in combination with the Application Suite X software (Leica Microsystems). Bodipy 493/503 was excited at 488 nm and emission was detected in the range between 510 and 560 nm. 3D stacks of the cardiomyocytes were imaged at a magnification of 63x and the corresponding intensity projections were analyzed with Fiji (ImageJ). The Trainable Weka Segmentation Plugin was used to differentiate between background, cardiomyocytes and lipid droplets. The results are presented as the ratio between the lipid droplet area and the cardiomyocyte area.

## Liquid chromatography mass spectrometry (LC-MS) for lipid analysis

Fresh frozen septa were homogenized in 0.1% ammonium acetate using a Tissuelyser (Qiagen, Germany). The homogenate was spiked with 10 μL of a stable isotope-labeled internal standard lipid mix (EquiSPLASH®, Avanti Polar Lipids, USA). Extraction was based on

Matyash et al. [72] Briefly 175 µL methanol (MeOH) and 625 µL *tert*-butyl methyl ether (MTBE) were added to the homogenate. After incubation at 4 °C for 1 h, 200 µL water was added. Phase separation was achieved by centrifugation (Microfuge® 20 R, Beckman Coulter, USA) for 10 min at 4 °C and 5000 g. The upper phase was collected, and the lower phase re-extracted as described above. The combined organic phases were dried in a vacuum concentrator (Concentrator 5301, Eppendorf, Germany). Samples were reconstituted in 50 µL MeOH/chloroform/isopropanol (iPrOH) (2:1:1) and stored at −80 °C. Samples were diluted by factor 100 in MeOH before measurement.

Lipids were analyzed using an UHPLC system (Vanquish Flex, Thermo Fisher Scientific, Germany) equipped with an Ascentis® Express C18 column (150 × 2.1 mm: 2.7 µm, 90 Å, Supelco, Germany) coupled to a HESI-orbital trapping mass spectrometer (Exploris 240, Thermo Fisher Scientific, Germany). The method used was based on Criscuolo et al. [73] with slight changes. In short, separation was achieved by a 36 min gradient of mobile phase A (MPA) (acetonitrile(ACN)/water, 1:1, v/v) and mobile phase B (MPB, i-PrOH/ACN/water, 85:10:5, v/v/v) both containing 5 mM $NH_4HCO_2$ and 0.1% formic acid at a flow rate of 300 µL/min, a column oven temperature of 50 °C, and an injection volume of 5 µL. Gradient: 0–20 min from 10–86% MPB, 20 – 22 min from 86–100% MPB, 22–27 min isocratic flow, 27–28 min from 100–10% MPB, 28–36 min isocratic flow. MS measurements were carried out in data dependent top 10 acquisition mode. Survey scans were performed at a resolution of 180,000 in a scan range of m/z 150–1200. For dd-MS2 a resolution of 15,000, minimum signal threshold of 5,000, isolation window of 2.0 m/z, and stepped HCD activation energies of NCE 20, 25, 30 were set. Spectra were analyzed using Lipostar2 (Molecular Discovery, Italy). Only lipids with confirmed lipid class by MS2 and matching isotopic patterns were used for analysis.

A heatmap of identified TAG, DAG and ceramides, with fatty acid components (palmitic acid [16:0], stearic acid [18:0], oleic acid [18:1]), which are ingredients of the used HFD (E15742-34; ssniff, Germany), was generated in Perseus (MPI Biochemistry, version 2.0.9.0) (Fig. S11, Supplementary Data 2).

## Transmission Electron Microscopy (TEM)

For TEM analysis, hearts were quickly removed and cut into smaller sections within 30 s post-explantation. These sections were immediately placed in primary fixative solution (2% formaldehyde, 2.5% glutaraldehyde, and 2 mM $CaCl_2$ in 0.15 M cacodylate buffer), 1 mm³ pieces were then transferred into a new fixative solution for 3 h at room temperature, washed three times with 0.15 M cacodylate buffer and then post-fixed and contrasted using a modified OTO protocol (2% osmium tetroxide + 1.5% potassium ferricyanide-(II), thiocarbohydrazide, 2% osmium tetroxide, 2% uranyl acetate) [74]. After dehydration in an ascending ethanol series, the samples were placed in propylene oxide (PO, 2 × 2 min), embedded in epoxy resin (Epon) with increasing resin concentration (Epon:PO 1:1, 3:1) and treated overnight in pure Epon, followed by a polymerization step in fresh Epon for 48 h at 60 °C. After cutting into 2 × 2 mm blocks (Leica EM TRIM), thin sections of approximately 70 nm were generated using an ultramicrotome (Ultracut E, Reichert Jung) and transferred to thin copper grids (75 mesh). Images were acquired using a Zeiss EM910 transmission electron microscope equipped with a tungsten cathode operating at 80 kV, along with ImageSP software (version 1.2.13.33 (x64), TRS Restlichtverstärkersysteme). Digital 16-bit images were captured with a CCD camera (TRS) at a resolution of 2048 × 2048 pixels and various magnifications. Images were post-processed using Fiji (Version 2.14.0/1.54 f) [75] and GNU Image Manipulation Program (GIMP, Version 2.10.34). Quantitative analysis of the mitochondria (number, area) was performed by using 24 randomly selected TEM images of each group, including 4-5 mice at a magnification of 10 K in Fiji. For statistical analysis and graph generation, Prism 10 software was used (GraphPad, San Diego, CA, USA).

## Quantitative real-time PCR

Total mRNA was isolated from murine frozen tissues using the miRNeasy Micro Kit (Qiagen, Hilden, Germany) following the manufacturer´s standard protocol. Reverse transcription was performed for 30 min at 42 °C using dNTP Mix (10 mM each, VWR, Radnor, PA, USA) and SuperScript II Reverse Transcriptase (Thermo Fisher Scientific, Waltham, MA, USA). qPCR was carried out on StepOnePlus (Applied Biosystems, Foster City, CA, USA) using Power Sybr® Green qPCR Master Mix (4367659, Applied Biosystems, Foster City, CA, USA) and specific primers: *Sirt1-F*: 5′-GTGTCATAGGCTAGGTGGTGA-3′; *Sirt1-R*: 5′-TCCTTTTGTGGGCGTGGAGG-3′; *Nmnat1-F*: 5′-GTGCCCAACTTGTGGAAGAT-3′; *Nmnat1-R*: 5′-CAGCACATCGGACTCGTAGA-3′; *Rpl32-F*: 5′-CGGAAACCCAGAGGCATTGA-3′ *Rpl32-R*: 5′-GGACCAGGAACTTGCGGAAG-3′; *Sod1-F*: 5′-AACCATCCACTTCGAGCAGA-3′; *Sod1-R*: 5′-TACTGATGGACGTGGAACCC-3′. Tomm70 and Sod2 mRNA expression was measured with TaqMan real-time PCR assay on StepOnePlus (Applied Biosystems, Foster City, CA, USA) using HotStarTaq DNA Polymerase (Qiagen, Hilden, Germany) according to the manufacturer's instructions. The primer sequences are: *Sod2-F*: 5′-GTGGTGGAGAACCCAAAGGAGA-3′; *Sod2-R*: 5′-TGAACCTTGGACTCCCACAGA-3′; *Sod2-P*: 5′-TTGCTGGAGGCTATCAAGCGTGA-3′; *Rpl32-F*: 5′-GCTGATGTGCAACAAATCTTA-3′; *Rpl32-R:* 5′-TCGGTTCTTAGAGGACACA-3′; *Rpl32-P*: 5′-TGTGAGCAATCTCAGCAC-3′. ΔCT was calculated related to the expression of the ribosomal protein L32 (*Rpl32*). $2^{-\Delta\Delta CT}$ was calculated reflecting expressions relative to the mean of the chow group.

## Mitochondrial copy number

Mitochondrial copy numbers were quantified using DNA by TaqMan real-time PCR assays. Total DNA was isolated from murine frozen tissues using the Pure Link Genomic DNA Kit (Life Technologies, Carlsbad, CA, USA) following the manufacturer´s standard protocol. RT-PCR was carried out on StepOnePlus (Applied Biosystems, Foster City, CA, USA) using HotStarTaq DNA Polymerase (Qiagen, Hilden, Germany) according to the manufacturer's instructions. Specific primer sequences and probes targeting mitochondrial DNA (mtDNA) or genomic DNA (gDNA) were used: mtDNA-F: 5′-TCGCAGCTACAGGAAAATCA-3′; mtDNA-R: 5′-TGAAACTGGTGTAGGGCCTT-3′; mtDNA-P: 5′-GCACAATTTGGCCTCCACCCA-3′; gDNA-F: 5′-CAGAGACAGCAAACATCAGA-3′; gDNA-R: 5′-CAGGGTGATGGAGAAGGA-3′; gDNA-P: 5′-CCCGACCCACGCCAGCAT-3′. $2^{-\Delta\Delta CT}$ was calculated reflecting expressions relative to the mean of the chow group.

## High-resolution respirometry

Cardiomyocytes were used for high-resolution respirometry directly after isolation or after 48 h culture as described above. The cardiomyocytes were collected after a percoll gradient centrifugation (see above section immunoblotting) and washed three times with 4 ml mitochondrial respiration buffer MiR05 (60101-01, Oroboros, Innsbruck, Austria). The oxygen consumption rate (OCR) of the cardiomyocytes was measured using a polarographic oxygen sensor (O2K-fluo, Oroboros Instrument, Innsbruck, Austria) and a substrate-uncoupler-inhibitor titration protocol (SUIT). Briefly, 10,000 cells were added to the respiration chambers containing 2 ml MiR05 buffer, at 37 °C. The cells were first permeabilized by addition of 25 µg/ml saponin (Sigma Aldrich), followed by sequential addition of substrates (Sigma Aldrich) and inhibitors (Sigma Aldrich) of the mitochondrial electron transport chain: 0.5 mM carnitine, 40 µM palmitoyl-CoA, 0.1 mM malate, 1.5 mM $Mg^{2+}$, 2.5 mM ADP, 5 mM pyruvate, 6 mM glutamate, 2 mM malate, 10 µM cytochrome c, 10 mM succinate, 2.5 µM oligomycin, 2 µM FCCP, 0.5 µM rotenone and 2.5 µM antimycin A.

## Statistical analysis

All data are shown as mean ± standard deviation (SD) unless otherwise indicated. The sample size is listed as N. Statistical differences were

determined using GraphPad Prism Version 10.0 for Windows. Data were tested for normality using the Kolmogorov-Smirnov test. For multiple independent groups, one-way analysis of variance (ANOVA) followed by Bonferroni's post-hoc test for parametric, and Kruskal-Wallis test followed by Dunn's multi-comparison test for non-parametric data were used, unless otherwise indicated. For a within-group comparison over time, a marginal linear mixed effect model with Bonferroni's post-hoc test was used. Fixed variables were time-point, treatment, and interaction, while the individual animal was included as a random variable. Unless otherwise stated, the significances are shown compared to the respective control data. In detail, $*p < 0.05$; $**p < 0.01$; $***p < 0.001$; $****p < 0.0001$. The exact $p$-values are provided in the source data file.

### Reporting summary
Further information on research design is available in the Nature Portfolio Reporting Summary linked to this article.

### Data availability
All data generated or analyzed during this study are included in this published article and its supplementary information files. Source data are provided in a source data file. The mass spectrometry proteomics data have been deposited to the ProteomeXchange Consortium via the PRIDE[76] partner repository with the dataset identifier PXD058165; ProteomeXchange title: Nitro-oleic acid in a HFpEF two-hit mouse model of high-fat diet and L-NAME. Source data are provided with this paper.

### Code availability
The custom code for evaluating pulmonary artery muscularisation is available at https://github.com/pLeminoq/vesseval, where it is published under an open source licence [https://doi.org/10.5281/zenodo.15052232]. The link includes installation and usage instructions.

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

## Acknowledgements

We thank Désirée Gerdes and André Grafe for expert technical assistance and Paul Steendijk for extensive advice regarding analysis of pressure-volume recordings. This work was supported by the Deutsche Forschungsgemeinschaft, Bonn, Germany (RU 1678/3-3 to V.R.; KL 2516/5-1 to A.K.; MU 4726/2-1 to M.M.; INST 213/973-1 FUGG), by the Deutsche Stiftung für Herzforschung, Frankfurt a.M., Germany (F/ 48/ 20 to M.M. and A.K.), by the Deutsche Diabetes Gesellschaft (DDG/21 to M.D.) and by FoRUM, Bochum, Germany (F991R-2021 to A.K. and V.R.; F1066-2022 to M.M. and A.K.), by NIH R01GM125944 (to F.J.S.) and NIH R61HL157069, R01HL064937 (to B.A.F.). The work conducted at the Leibniz-Institut für Analytische Wissenschaften – ISAS – e.V. was supported by the "Ministerium für Kultur und Wissenschaft des Landes Nordrhein-Westfalen", "Senatsverwaltung für Wissenschaft, Gesundheit und Pflege Berlin", and by the "Bundesministerium für Bildung und Forschung (BMBF)" (for F.-L.H, J.D., E.T., K.L., S.H.).

## Author contributions

All authors made substantial contributions to the conception or design of the study; the acquisition, analysis, or interpretation of data; or drafting or revising the paper. All authors approved the paper. All authors agree to be personally accountable for individual contributions and to ensure that questions related to the accuracy or integrity of any part of the work are appropriately investigated, resolved, and the resolution documented in the literature. M.M., U.S., A.K. conception and design of research; M.M.; T.S., C.W., T.M., T.P.; E.D., J.HW, F.L.H, T.J.S., L.A.L., L.L., E.T.V., J.H., S.L., J.D. U.S. performed experiments; M.M., T.S., C.W., T.M., T.P., J.HW., F.L.H., T.J.S., L.A.L., L.L., J.H., S.L., E.T., J.D., T.H., J.C.R., U.S., A.K. analyzed data; M.M., T.S., J.HW., S.H., B.S., M.D., E.T., J.D., K.L., J.C.R., U.S., A.K. interpreted results of experiments; M.M., T.S., J.HW., J.H., M.D., U.S., A.K. prepared figures; M.M., A.K. drafted the manuscript; S.H., M.D.; K.L., V.S., F.J.S., B.A.F., V.R. edited and revised the manuscript.

## Funding

## Competing interests

BAF acknowledges an interest in Creegh Pharmaceuticals, Inc., and FJS acknowledges an interest in Creegh Pharmaceuticals, Inc. and Furanica, Inc. The other authors declare no conflict of interest.

## Additional information

¹Clinic for General and Interventional Cardiology/ Angiology, Herz- und Diabeteszentrum NRW, Ruhr-Universität Bochum, Bad Oeynhausen, Germany. ²Agnes Wittenborg Institute for Translational Cardiovascular Research (AWIHK), Herz- und Diabeteszentrum NRW, Ruhr-Universität Bochum, Bad Oeynhausen, Germany. ³Technology Platform Genomics, Center for Biotechnology (CeBiTec), Bielefeld University, Bielefeld, Germany. ⁴Medical Imaging Center (MIC), Electron Microscopy Medical Analysis – Core Facility (EMMACF), Med. Fakultät, Ruhr-Universität Bochum, Bochum, Germany. ⁵Leibniz-Institut für Analytische Wissenschaften-ISAS e.V., Dortmund, Germany. ⁶Faculty of Chemistry, University of Duisburg-Essen, 45141 Essen, Germany. ⁷Diabetescenter, Herz- und Diabeteszentrum NRW, Ruhr-Universität Bochum, Bad Oeynhausen, Germany. ⁸Institute of Pharmaceutical and Medicinal Chemistry, University of Münster, Münster, Germany. ⁹Institute of Pharmacology and Toxicology, University of Würzburg, Würzburg, Germany. ¹⁰Institute for Radiology, Nuclear Medicine and Molecular Imaging, Herz- und Diabeteszentrum NRW, Ruhr-Universität Bochum, Bad Oeynhausen, Germany. ¹¹Department of Translational Science Universitätsklinikum, DZHI, Würzburg, Germany. ¹²Department of Pharmacology and Chemical Biology, University of Pittsburgh, Pittsburgh, PA, USA. ¹³These authors contributed equally: Uwe Schlomann, Anna Klinke. ✉e-mail: aklinke@hdz-nrw.de

