## [Peer Review File · Nature Communications]

Nitro-oleic acid enhances mitochondrial metabolism and ameliorates Heart Failure with Preserved Ejection Fraction in mice

Corresponding Author: Dr Anna Klinke

Version 0:

Reviewer comments:

Reviewer #1

(Remarks to the Author)

The study by Mueller and colleagues investigates the effects of nitro-oleic acid (NO₂-OA) in a mouse model of heart failure with preserved ejection fraction (HFpEF), using high fat diet (HF) and L-NAME in C57/BL6N mice. NO₂-OA is an electrophilic molecule formed by reactions of nitrogen oxides, particularly nitrogen dioxide, with fatty acids containing conjugated double bonds. Previous studies, also by this group, revealed that NO₂-OA improves glucose tolerance and adipocyte function. Here, after 11 weeks of treatment with HFD/L-NAME, an additional 2-4 weeks of treatment with NO₂-OA improved diastolic function and global longitudinal strain in HFpEF mice. Proteomic analysis followed by Western blot revealed that NO₂-OA upregulated mitochondrial proteins in HFpEF mice. In addition, upregulation of AMPK and downstream signaling as well as mitochondrial biogenesis were boosted. In isolated cardiac myocytes cultured for 48 hours in vitro, oxidative phosphorylation was increased. The authors conclude that NO₂-OA improves mitochondrial integrity and thereby, diastolic function in HFpEF.

The study addresses a clinically important and scientifically interesting topic that is at the same time very translational, since a) HFpEF has few effective treatments and b) NO₂-OA is in principal clinically safe and has been applied in clinical studies already. The manuscript is very well written and the results clearly presented. Limitations are addressed adequately. While the proteomic data look very intriguing, a better phenotypisation of cardiac function in the HFpEF model and in particular, more functional assays and/or in vivo measurements would be required to better delineate what is happening on the level of mitochondria and substrate utilization.

- 1) In vivo Phenotypisation: It would be good to have additional markers for heart failure in the mouse model, such as cardiac expression of ANP and BNP. An ideal way to determine diastolic function would be pressure-volume analysis, but echo plus biomarkers as indicated are presumably sufficient for the purpose of the study.
- 2) In addition to cardiac function, histology would be good to discriminate whether NO₂-OA affects cardiac myocyte hypertrophy and/or fibrosis, relevant factors for diastolic function.
- 3) The proteomic data are intriguing. Considering that the by far strongest effect is the one by NO₂-OA in HFpEF mice, while the changes induced by HFD/L-NAME are rather modest, it would be important to understand whether this effect of NO₂-OA is similarly strong in healthy animals. While I understand that redoing proteomic analysis is extensive, alternatively, Western Blot experiments for some of the key findings (such as the ones reported in Figure 4c and d) would be of interest.
- 4) It is a bit surprising to see that PDK4 is further upregulated by NO₂-OA, as is p-PDH, in light of the other observations that mitochondrial biogenesis and proteins involved in oxidative phosphorylation are also rather upregulated. Together with the strong effects on FA uptake and oxidation proteins, this suggests that FA oxidation is turned on even stronger. It would be helpful to determine substrate utilization in vivo, such as by PET (FDG and FTO) to determine FA and glucose uptake/oxidation.
- 5) Mitochondrial respiration measurements: Determining mitochondrial respiration in isolated adult cardiac myocytes, in particular after 48 hours of cultivation, plus with quite artificial conditions, is problematic. This needs to be validated by more reliable assays. The easiest would be to isolate cardiac mitochondria from hearts and determine oxygen consumption with a Clark electrode / Oroboros system, employing various substrates (fatty acids, pyruvate/malate, glutamate/malate).
- 6) As mentioned in your own limitations section, elucidating whether improvement of diastolic (and potentially, also systolic)

function is related to cardiac myocyte function (excitation-contraction coupling), analyses of these would be helpful, but not essentially required.

Reviewer #2

(Remarks to the Author)

This paper has great potential as it investigates effect of Nitro-oleic acid on one solid model of HFpEF and provides interesting data that is thought provoking. However there are number of challenges mainly around lack of transparency which should be easy to include

Major concerns

1. The authors must provide much more clarity around how the mass spectrometry (MS) data was handled. For the peptide level data and include which library was used to search the data, and which modification were allowed etc. For the step where the peptide level data is used to infer the protein changes, did the authors sum/include common peptide which have the same amino acid sequence to more than one protein as oppose to inclusion of only proteotypic peptides; were isoforms only assigned if a peptide has an unique amino acid sequence for that isoform etc and how cutoff for FDR and log₁₀ FC (note log₂ FC is normally used) and whether FDR/p value was adjusted for multiple variables. In addition, how did the authors deal with missing data such as if a protein seen in only subset of tissue or myocytes or changed in only subset of these animals/cultures. Please expand methods and include all this information including what library was used to search the data, if modification were allowed etc.
2. The group must report all peptide and protein level data from their mass spectrometry for each animal and experiment. Coupled with lack of detail on how the data was processed makes it impossible to judge the quality of the data and thus, interpretation is correct. This data can go in as set of online supplement tables. The MS data need or should be uploaded to an open source data repository like MASSIVE or Pride.
3. In figures S7, S8 and S9 figures of the total protein (SPL red) shows that for the tissue extracts the high MW cardiac proteins are missing (like myosin heavy chain, myosin binding protein C and titin). This is suggestive of incomplete solubilization or precipitation of the proteins. Without the MS data to support the work, one cannot check what maybe going on, as these dominant proteins must be present if full extraction or solubilization has occurred. It is also interesting/concerning that the total protein stain of the gel of the cardiomyocytes shown in S10 do have these dominant high MW proteins. Without knowing if full extract (and complete digestion) of the tissue has occurred then again, it is not possible ensure that interpretation is correct.

Minor but important

1. Clearly state sex of animals (I assume they are all male).
2. For figure 1A it may be helpful to include when NO₂-OA/OA was given and when samples were harvested.
3. For figure 3. It is important add protein names for heat maps. It is also essential to include a table of all differentially expressed proteins for each comparison with their log change and adjusted FDR/p value.

Reviewer #3

(Remarks to the Author)

The authors investigated the effects of nitro-oleic acid therapy in a popular murine model of cardiometabolic HFpEF. The authors demonstrate a cardiac phenotype of HFpEF following 11 weeks of L-NAME + high fat diet feeding to the mice. The authors provide some evidence that nitro-oleic acid therapy attenuates diastolic dysfunction in the 2-hit HFpEF model. The study addresses a significant clinical issue and provides some data to demonstrate the potential benefit of a novel therapeutic agent. The authors provide extensive data to show that NO₂-OA improves cardiac mitochondrial function and overall metabolic function.

Overall, the studies have been performed in a careful manner and the data do support the author's conclusion that NO₂-OA exerts beneficial actions in this preclinical HFpEF model.

I have a few comments and concerns regarding the paper.

- (1) The authors provide data that NO₂-OA improves LV diastolic function. The entire study rests on 2-D ECHO data. The authors should provide data for exercise capacity (treadmill exercise) since this is essentially the most important outcome in patients that suffer from HFpEF in addition to overall quality of life. ECHO data alone is not sufficient.
- (2) Data for HF severity biomarkers and for inflammation and oxidative stress should be provided. The authors should evaluate BNP and hsCRP, etc.
- (3) The authors report on HFpEF in male mice only. The authors should also evaluate NO₂-OA for efficacy only in female mice.
- (4) The authors should provide data for left ventricular pressures and for LV wall thickness and chamber dimensions.
- (5) One of the major discrepancies in the HFpEF field is the report by Schiattarella and Hill that the primary driver of HFpEF in this same model is nitrosative stress resulting from activation of iNOS and overproduction of NO. This is in sharp contrast

to preclinical and clinical reports of attenuated NO bioavailability and impaired endothelial dysfunction leading to coronary microvascular dysfunction and worsening HFpEF.

The authors must evaluate how NO₂-OA impacts nitrosative stress.

NO₂-OA should increase NO bioavailability and the authors should measure cardiac and global NO levels in HFpEF Control and HFpEF + NO₂-OA mouse groups.

The authors should also measure blood pressure.

This is a critically important issue that needs to be addressed.

(6) The authors fail to fully address the limitations of this preclinical model in which they induce obesity, hypertension, and HFpEF in an extremely short time. There are several other weaknesses to this model.

Version 1:

Reviewer comments:

Reviewer #1

(Remarks to the Author)

The authors have addressed all my concerns. I have no further questions.

(Remarks on code availability)

Reviewer #2

(Remarks to the Author)

Many of my concerns have been addressed however there are still some challenges remaining.

Major issues

1. MS methods have been expanded to include some of the needed information however there remains some important clarifications needed. Please clarify the following.

- “only unique peptides were included in the analysis” means that peptides with amino acid sequence that can be assigned to a single unique protein (prototypic peptide) was used. This may not be the case (see point 2). As well confirm that only the prototypic peptides were used in the protein identification and quantification.
- “Only proteins with valid values in all replicates of each group were considered” means the protein had to be present in all biological replicate in all sample or do you mean in each experimental group?
- “Proteins considered as potential contaminants were filtered out” means those that were from blood (e.g., hemoglobin?). Be specific.
- The authors set the FDR at peptide and protein level at 0.05, which is high (typically 0.01). Please add this to the limitation section as it is due, in part, to the MS instrument used.

2A. Very importantly, the authors have NOT included the peptide level data (both sequence and modified residues are needed) making it impossible to determine if the protein identification and quantification are correct.

Data for example in supplement table 1 (485599_1_data_set_9946709_sncr6n as well as in other tables such as 485599_1_data_set_9946716-sncr93) suggests there is an issue with protein identification being exclusively based on proteotypic peptides is illustrated by alpha-actin isoform assignments. Alpha-actin is extremely highly conserved (>95%) between the three isoforms expressed in different muscle types. In table 1, only 2 isoforms of alpha actin isoforms are listed and neither is the cardiac isoform, the isoform that should be present. The data reported in Table 1 for alpha-actin is i) uniprot ID P68134, gene name Acta1 (protein name is ACTS) which is actin, alpha skeletal muscle and ii) uniprot ID P62737, gene name Acta2 (protein name ACTA) which is actin, aortic smooth muscle. The actin isoform that should be in cardiac muscle is uniprot ID P68033 gene name Actc1 (protein name ACTC) which is actin alpha cardiac muscle. As alpha actin is highly conserved, this suggests that incorrect identification has occurred. This may be a larger issue as P05977 myosin light chain 1/3 skeletal muscle while cardiac is myosin light chain 2 and 3. The authors need to ensure that truly prototypic peptides are being used for identification and quantification.

2B. The authors have provided the requested protein data.

- Supplement table 1. Remove columns A, B and C (e.g., master protein column) or define their meaning, add in Uniprot identifier as these define the protein, add in the protein name (not just the gene name), define each column labeled as peptide (e.g., column J is labeled # peptides includes those peptides that are shared between more than one protein as well as those that are unique to a single protein (prototypic peptides). How column J differs from the number of peptides listed in columns R and S is not clear. In the methods, the authors stated only unique proteolytic peptides (column L) are used for protein. Therefore, consider removing the other columns. As well, confirm what are the values listed for each sample (is this average or total sum of the peak area or of the spectral counts for each proteotypic peptide. How these values are used for the protein quantification in the other tables must be clarified including if the quantification is based on unique prototypic peptides. If not, these data need to be redone using quantitation only the prototypic peptides.
- Supplement table 2. For each Tab, reduce decimal points to two or three spaces after the decimal place for -log P value

and difference. Define what is meant as it is not clear how this was determined, define or remove columns F to I and define animal experiment group of each sample in columns M- AI.

• Supplement Table 3, 4 and 5 (protein and lipid). For each Tab, reduce decimal points to two-three spaces after the decimal place.

3. Thank you for adding in the information to the reviewers comments about high MW proteins for mass spectrometry (Figure 2.1) and the additional gel images (figure 2.2). These were very informative and must be included in the online supplement as this data confirms that there is an under representation in the quantity of the high MW proteins.

Although detected by MS the protein ratio of MYHC6, MYBPC and titin the quantitation of these proteins do not seem to match up with respect to the lower MW proteins. This can be shown by plotting the dynamic range (total intensity of the prototypic peptides) across the proteome quantify and show where the representative high and low MW protein (e.g., cardiac TnT, cardiac TnI, MYH6, Titin, cardiac alpha actin). This issue is better illustrated by the gel stained images which clearly shows low amount of staining of the high MW proteins especially compared to alpha cardiac actin (the dark 42K band (which is not identified by MS as discussed above)) in Figure 2.2 A and B added to the response to reviewers (note: it is not clear if C is overexposed as no linear range was of the westerns was included). This under representation of the high MW protein is also seen in the gels for tissue (Figure 12.a, 13a, 14a) in the online supplement of the main manuscript (compare to the cardiomyocytes Figure 15) suggesting this is a wider spread issue. This needs to be addressed and should be included in a limitation section.

Minor issues

1. Thank you for including female animals.
2. Thank you for expanding the information.
3. Thank you for expanding the information.

(Remarks on code availability)

Reviewer #3

(Remarks to the Author)

No Further Comments for the Authors. The authors have gone above and beyond the questions and comments posed by the Reviewers.

The Authors have performed an extraordinary number of experiments, assays, and measurements to address all of the comments and suggestions provided in the previous review.

The paper is substantially improved and provides new and important information on the pathobiology of HFpEF, nitro-oleic acid, and HFpEF therapeutics.

The authors should be commended for an outstanding job.

(Remarks on code availability)

Version 2:

Reviewer comments:

Reviewer #2

(Remarks to the Author)

Thank you for the careful responses to the remaining queries. The changes made in the manuscript and online supplements have helped with transparency and confidence. This remains an exciting and important study.

(Remarks on code availability)

I did not run it but did take a look through the app and it was fine. I did notice that the demo dataset had an issue "Sorry about that, but we can't show files that are this big right now." and I am running the most recent windows. But, it could be something on my end.

The read me files are fine and transparent.

Response to Reviewer

Reviewer #1:

The study by Mueller and colleagues investigates the effects of nitro-oleic acid (NO₂-OA) in a mouse model of heart failure with preserved ejection fraction (HFpEF), using high fat diet (HFD) and L-NAME in C57/BL6N mice. NO₂-OA is an electrophilic molecule formed by reactions of nitrogen oxides, particularly nitrogen dioxide, with fatty acids containing conjugated double bonds. Previous studies, also by this group, revealed that NO₂-OA improves glucose tolerance and adipocyte function. Here, after 11 weeks of treatment with HFD/L-NAME, an additional 2-4 weeks of treatment with NO₂-OA improved diastolic function and global longitudinal strain in HFpEF mice. Proteomic analysis followed by Western blot revealed that NO₂-OA upregulated mitochondrial proteins in HFpEF mice. In addition, upregulation of AMPK and downstream signaling as well as mitochondrial biogenesis were boosted. In isolated cardiac myocytes cultured for 48 hours in vitro, oxidative phosphorylation was increased. The authors conclude that NO₂-OA improves mitochondrial integrity and thereby, diastolic function in HFpEF.

The study addresses a clinically important and scientifically interesting topic that is at the same time very translational, since a) HFpEF has few effective treatments and b) NO₂-OA is in principal clinically safe and has been applied in clinical studies already. The manuscript is very well written and the results clearly presented. Limitations are addressed adequately. While the proteomic data look very intriguing, a better phenotypisation of cardiac function in the HFpEF model and in particular, more functional assays and/or in vivo measurements would be required to better delineate what is happening on the level of mitochondria and substrate utilization.

We thank the reviewer for the positive perception of our work. To strengthen our functional data, we performed left ventricular pressure-volume loop analyses and treadmill tests to explore the effects of NO₂-OA on cardiac hemodynamics and exercise capacity in the HFpEF mouse model. In addition, high-resolution respirometry was done in isolated cardiomyocytes of HFpEF mice with and without NO₂-OA treatment to decipher alterations in substrate utilization and mitochondrial function.

1) In vivo Phenotypisation: It would be good to have additional markers for heart failure in the mouse model, such as cardiac expression of ANP and BNP. An ideal way to determine diastolic function would be pressure-volume analysis, but echo plus biomarkers as indicated are presumably sufficient for the purpose of the study.

We agree that additional functional data would strengthen the robustness of our findings. We therefore created an additional set of chow, HFpEF/vehicle (HFD+L-NAME) and HFpEF/NO₂-OA (HFD+L-NAME NO₂-OA) mice (N=19/14/18). To make sure that the cardiac phenotype is similar to what we observed before (initially we used mice from Charles River, this time mice were obtained from Janvier Labs), we performed echocardiography again, which confirmed our previous findings (Response Fig. 1.1). The mice were randomly assigned to either final hemodynamic measurements or cardiomyocyte isolation for respirometry. The invasive pressure volume analyses revealed a significantly increased LV end-diastolic pressure in vehicle-treated HFpEF mice but not in NO₂-OA-treated mice (Resp. Fig. 1.2 a, Manuscript Fig. 1g). The chamber stiffness index β_w , which is the slope of the end-diastolic pressure-volume relation multiplied by wall thickness¹, corroborates the mitigation of diastolic stiffness by NO₂-OA treatment as seen by echocardiography (Resp. Fig. 1.2 b, MS. Fig. 1h).

Response Figure 1.1

Resp. Fig. 1.1: Mice received high-fat diet and the endothelial nitric oxide synthase inhibitor L-NAME (HFD+L-NAME) or chow diet (Chow) until induction of the HFpEF phenotype (11 to 15 weeks). Treatment with vehicle or nitro-oleic acid (NO₂-OA) was conducted for the last 4 weeks. Echocardiography was performed at the final time point to confirm our previous finding in the “Initial cohort” (a) (N=11/10/11) and “Revision cohort” (b) (N=19/14/18). Statistical significance was calculated by One-way ANOVA followed by Bonferroni’s post-hoc test except for compliance index in **b** Kruskal-Wallis-test followed by Dunn’s post-hoc test was used. p: *<0.05, **<0.01, ***<0.001, ****<0.0001.

Response Figure 1.2

Resp. Fig. 1.2: Mice were subjected to hemodynamic measurements using an open chest approach. a) Left-ventricular end-diastolic pressure (LV EDP) and end-systolic pressure (LV ESP). b) Chamber stiffness index (β_w). PV-loop analysis was performed with Circlab (v12.3) using a single-beat estimation. Statistical significance was calculated using One-way ANOVA with Bonferroni’s post-hoc test (N=11/7/6). p: *<0.05, **<0.01, ***<0.001.

Quantification of BNP protein levels in isolated cardiomyocytes also supports the phenotypic characteristics (Resp. Fig. 1.3). In addition, we determined exercise capacity in all animals revealing that work rate was significantly smaller in vehicle-treated but not NO₂-OA-treated HFpEF mice compared to chow animals (Resp. Fig. 1.4a, MS. Fig. 1i). An extended evaluation of echocardiography data confirmed the protective effects of NO₂-OA treatment in HFpEF mice, showing a significant reduction of left atrial (LA) area (Resp. Fig. 1.4b, MS. Fig. 1j). and

pulmonary vascular resistance (Resp. Fig. 1.4c) compared to vehicle. In line, muscularization of pulmonary arteries (PA) was significantly increased in HFD+L-NAME mice compared to chow animals. NO₂-OA treatment blunted PA muscularization as assessed by immunofluorescence staining of smooth muscle actin on murine lung slices (Resp. Fig. 1.4d, MS. Fig. 1k).

Response Figure 1.3

Resp. Fig. 1.3: Protein level of brain-natriuretic peptide (BNP) was detected in isolated murine cardiomyocytes of chow, vehicle-treated HFD+L-NAME and NO₂-OA-treated HFD+L-NAME mice. The relative BNP level significant negatively correlated with early diastolic mitral annulus velocity (e') and pulmonary artery volume time integral (PA VTI). Statistical significance was calculated by One-way ANOVA followed by Bonferroni's post-hoc test and Pearson's correlation ($N=8/6/5$).

Response Figure 1.4

Resp. Fig. 1.4: a) Work rate was calculated from graded exercise test including running speed, incline and body weight ($N=18/14/17$). b, c) Left atrial area (LA Area, b) and pulmonary vascular resistance (TR peak velocity/PA VTI, c) detected by echocardiography ($N=19/14/18$). d) Muscularization of pulmonary arteries was assessed by immunofluorescence staining of smooth muscle actin on murine lung slices. Vessels with a diameter ranging from 10-110 μm were analysed ($N=9/8/9$). Statistical significance was calculated by One-way ANOVA followed by Bonferroni's post-hoc test. p : * <0.05 , ** <0.01 , *** <0.001 .

2) In addition to cardiac function, histology would be good to discriminate whether NO₂-OA affects cardiac myocyte hypertrophy and/or fibrosis, relevant factors for diastolic function.

Analysis of interstitial collagen by picrosirius red stain in LV sections showed increased fibrosis in both HFpEF groups compared to chow mice (Resp. Fig. 1.5a, MS. Fig. S1a). In addition, LV hypertrophy was significantly increased in HFD+L-NAME mice compared to chow as reflected by increased LV posterior wall thickness (Resp. Fig. 1.1, MS Fig. 1b) and cardiomyocyte diameter assessed by pulse area analysis-based volumetry of isolated cardiomyocytes, which was performed before respirometry (Resp. Fig. 1.5 b, MS Fig. S2b).

Response Figure 1.5

Resp. Fig. 1.5: a) Representative images (scale bar 200 µm) of LV sections stained with picrosirius red and analysis of fibrosis grades (N=10/10/7). Images were scored based on the amount of collagen deposition. b) Mean cell diameter of isolated cardiomyocytes assessed by pulse area analysis-based volumetry (N=8/6/5). Statistical significance was calculated by Fisher exact test for a. * presents analysis of chow vs. HFD+L-NAME vehicle and # analysis of HFD+L-NAME vehicle vs. NO₂-OA group. One-way ANOVA with Bonferroni's post-hoc test was used for b. p: ***<0.001.

3) The proteomic data are intriguing. Considering that the by far strongest effect is the one by NO₂-OA in HFpEF mice, while the changes induced by HFD/L-NAME are rather modest, it would be important to understand whether this effect of NO₂-OA is similarly strong in healthy animals. While I understand that redoing proteomic analysis is extensive, alternatively, Western Blot experiments for some of the key findings (such as the ones reported in Figure 4c and d) would be of interest.

We absolutely agree that a deep understanding of the effect of NO₂-OA is desirable. Anyhow, it has not been elucidated well, how NO₂-OA's effects on certain signaling pathways differ between health and disease in other models. We now administered NO₂-OA with the same dose and duration as in HFpEF mice in healthy wildtype animals (Resp. Fig. 1.6). After 4 weeks we isolated cardiomyocytes from these mice and performed proteome analysis, immunoblot (IB), immunofluorescence staining (IF) and respirometry. In the proteome analysis of isolated cardiomyocytes, we identified 995 proteins in total. Comparing NO₂-OA-treated with untreated mice, we found 34 proteins with significantly higher protein abundances (3.4 %) and 9 proteins with significantly lower protein level (0.1 %) upon NO₂-OA treatment (Resp. Fig. 1.6 a). According to the mitoCarta data base², 20.9 % of the differently abundant proteins were mitochondrial proteins (9 out of 43 proteins) (Resp. Fig. 1.6 a, red marked proteins). Compared to the proteome analysis derived from LV tissue of HFpEF mice, the number of regulated mitochondrial proteins as well as the level of regulation were much smaller (-log₂(fold change): HFD+L-NAME NO₂-OA tissue -2.75-2.89; WT NO₂-OA cardiomyocytes -2.49-0.97). However, we found a significant activation of AMPK in NO₂-OA-treated cardiomyocytes by immunoblot similar to NO₂-OA-treated HFpEF heart tissue (Resp. Fig. 1.6b). In addition, immunoblot analysis of mitochondrial marker proteins, COX4 and HSP60, as well as immunofluorescence staining of TOM70 on isolated cardiomyocytes showed no differences between untreated and NO₂-OA-treated cardiomyocytes (Resp. Fig. 1.6c, d). Interestingly, Oroboros measurements revealed significantly enhanced ATPase-linked respiration and respiratory control rate, pointing towards improved coupling efficiency (Resp. Fig. 6e). We infer from these findings,

that the AMPK signaling pathway is induced also in healthy animals. But given that energy demands are not increased, there is no need for augmentation of mitochondrial numbers. Therefore, mitochondrial dynamics might dampen this induction as a resource sparing mechanism of the healthy cardiomyocyte. Of note, we did not check alterations on posttranslational level nor changes in mitochondrial quality control or mitochondrial fission and fusion dynamics in healthy $\text{NO}_2\text{-OA}$ treated animals. As we have not fully unraveled these mechanisms in healthy mice, we do not include these data in the revised manuscript. Further analyses are subject of ongoing experiments.

Response Figure 1.6

Resp. Fig. 1.6: Healthy mice (Ctrl) were treated with vehicle (Ctrl vehicle) or $\text{NO}_2\text{-OA}$ (Ctrl $\text{NO}_2\text{-OA}$) for 4 weeks. Cardiomyocytes were isolated. Proteome analyses, immunoblot (IB), immunofluorescence staining (IF) and respirometry was performed (N=6/6; unpublished data). a) Heatmap of proteins with significantly different abundances in $\text{NO}_2\text{-OA}$ -treated compared to vehicle-treated mice ($p \geq -\log_{10}(0.05)$; $-\log_2(1.5) < \text{fold change} < \log_2(1.5)$). Mitochondrial proteins are marked red. b) AMPK phosphorylation level relative to total AMPK level. c) Protein level of COX4 and HSP60, relative to total protein. d) Immunofluorescence signal intensity of TOM70. e) High resolution respirometry of permeabilized cardiomyocytes under controlled substrate supply. Complex I-, complex-II-linked respiration, absolute oxidative phosphorylation (OXPHOS), ATPase-linked respiration and respiratory control rate are extracted from oxygen consumption flux traces. Statistical significance was calculated by unpaired Student's t-test. p: * <0.05 .

4) It is a bit surprising to see that PDK4 is further upregulated by $\text{NO}_2\text{-OA}$, as is p-PDH, in light of the other observations that mitochondrial biogenesis and proteins involved in oxidative phosphorylation are also rather upregulated. Together with the stoing effects on FA uptake and oxidation proteins, this suggests that FA oxidation is turned on even stronger. It would be

helpful to determine substrate utilization in vivo, such as by PET (FDG and FTO) to determine FA and glucose uptake/oxidation.

We thank the reviewer for this notion. Given that PDK4 and PDH are expressed solely in mitochondria, we normalized protein levels to COX4 expression (representative for mitochondrial amount) in heart homogenates, which discloses that PDK4 expression and PDH inhibition per mitochondrial protein amount is increased in HFpEF mice compared to chow mice, but the increase is less pronounced in NO₂-OA-treated vs. vehicle-treated animals (Resp. Fig. 1.7a, MS. Fig. 2g, h). This means that the inhibition of GO is diminished instead of enhanced by NO₂-OA, which is in accordance with improved glucose tolerance (MS. Fig. 2f). Nevertheless, respirometry points towards the fact that FAO is enhanced predominantly in NO₂-OA-treated, not in vehicle-treated HFpEF mice (see response to comment 5).

Response Figure 1.7

- Chow ▼ HFD+L-NAME vehicle ♦ HFD+L-NAME NO₂-OA

Resp. Fig. 1.7: Protein abundance of mitochondrial pyruvate dehydrogenase kinase 4 (PDK4) normalized to cytochrome c oxidase subunit 4 (COX4) measured liquid chromatography – mass spectrometry (left) or immunoblot (middle). Pyruvate dehydrogenase (PDH) phosphorylation level normalized to COX4 protein level. Statistical significance was calculated by One-way ANOVA followed by Bonferroni's post-hoc test for PDK4/COX4 and by Kruskal-Wallis test followed by Dunn's post-hoc test for pPDH/COX4 (N=9/8/6). p: *<0.05, **<0.01, ****<0.0001

5) Mitochondrial respiration measurements: Determining mitochondrial respiration in isolated adult cardiac myocytes, in particular after 48 hours of cultivation, plus with quite artificial conditions, is problematic. This needs to be validated by more reliable assays. The easiest would be to isolate cardiac mitochondria from hearts and determine oxygen consumption with a Clark electrode / Oroboros system, employing various substrates (fatty acids, pyruvate/malate, glutamate/malate).

We followed the reviewer's suggestion and performed respirometry using the Oroboros system with freshly isolated cardiomyocytes from chow and HFpEF mice. We chose a protocol using first carnitine and palmitoyl-CoA, and secondly, pyruvate, glutamate, malate and succinate as substrates to receive information on β -oxidation, complex activities and OXPHOS. In contrast to healthy mice treated with NO₂-OA, but in accordance with the metabolically stressed cardiomyocytes, we observed a significant increase in OXPHOS in NO₂-OA-treated HFpEF mice. Furthermore, complex II activity and FAO were increased (Resp. Fig. 1.8, MS. Fig. 5a), which might be explained by the marked augmentation of mitochondrial protein amounts and numbers.

In addition, we performed transmission electron microscopy (TEM) in LV tissue of the three experimental groups (N=4/5/4). In accordance with our findings showing increased mitochondrial protein levels and enhanced mitochondrial copy number (MS. Fig. 3) in NO₂-OA-

treated HFpEF mice, we found a significantly increased number of mitochondria in HFD+L-NAME NO₂-OA mice compared to HFD+L-NAME mice as well as control animals (Resp. Fig. 1.8b, c, MS. Fig. 3e, f). The number of mitochondria showed a significant correlation with diastolic function (IVRT), indicating that increased number is related to improved function (Resp. Fig. 1.8d, MS Fig. 3g). Whereas mitochondria in control mouse heart tissue were intact, HFpEF mice showed cristae fragmentation and swollen mitochondria. Nevertheless, we did not observe decreased OXPHOS or ATPase activity in vehicle-treated HFpEF mice, nor significantly increased FAO, as it may be expected from mitochondrial architecture disruption and the general notion of decreased GO and increased FAO in metabolic HFpEF. Thus, our findings suggest, that under conditions of controlled substrate supply to the mitochondria (due to cell membrane permeabilization during the measurement), and without increased workload during respirometry, metabolic changes are not apparent. We conclude that NO₂-OA might induce glucose and fatty acid transport into the cardiomyocyte as indicated by significantly increased GLUT4, CD36 and CPT1B protein levels after NO₂-OA treatment in HFpEF mice (MS. Fig. 6b). The significantly increased number of mitochondria and potentially improved dynamics mediated by activation of the AMPK signaling pathway leads to enhanced oxidative phosphorylation. Enhanced metabolic rates, which offset lipid oversupply, might compensate energy deficits in metabolic HFpEF. We elaborated on this hypothesis further in the revised discussion section (ll. 686-696).

Response Figure 1.8

Figure 1.8: a) High resolution respirometry was performed in isolated, permeabilized cardiomyocytes with a protocol using carnitine, palmitoyl-CoA, pyruvate, glutamate and malate (N=8/6/5). b) Representative TEM images of murine LV tissue (N=5 animals per group) at magnifications (from top to bottom) of 5K (scale bar 5 μ m), 10K (scale bar 2 μ m), and 31.5K (scale bar 1 μ m). sc: sarcomere structure; ld: lipid droplet; mt: mitochondria with intact cristae structure; white arrow: early cristae fragmentation; white asterisk: cristae adhesion; black asterisk: swollen and ruptured mitochondria. c) Quantitative analysis of number of mitochondria normalized to mitochondrial area (24 images per group from N=4/5/4 animals). d) Correlation of isovolumetric relaxation time and number of mitochondria

per mitochondrial area. Statistical significance was calculated by One-way ANOVA followed by Bonferroni's post-hoc test or Pearson Correlation for d. $p: * < 0.05$.

(6) As mentioned in your own limitations section, elucidating whether improvement of diastolic (and potentially, also systolic) function is related to cardiac myocyte function (excitation-contraction coupling), analyses of these would be helpful, but not essentially required.

Although we have now performed additional experiments for which we isolated cardiomyocytes from chow and HFpEF mice, we analyzed respiration, but not excitation contraction coupling in these cells. However, data from proteomic analysis point towards effects of NO₂-OA on Ca²⁺ dynamics, i.e., significantly increased abundance of sarcoplasmic Ca²⁺ transporters: ryanodine receptor (RYR2) and sarcoplasmic/endoplasmic reticulum calcium ATPase (SERCA) (Resp. Fig. 1.9).

Response Figure 1.9

Resp. Fig. 1.9: Protein level of RYR2 and SERCA2 detected by liquid chromatography – mass spectrometry. Statistical significance was calculated by One-way ANOVA followed by Bonferroni's post-hoc test (N=9/8/6). $p: * < 0.05$, $** < 0.01$.

Reviewer #2 (Remarks to the Author):

This paper has great potential as it investigates effect of Nitro-oleic acid on one solid model of HFpEF and provides interesting data that is thought provoking. However, there are number of challenges mainly around lack of transparency which should be easy to include.

We thank the reviewer for his careful reading and important comments, which made us to re-analyse the raw LC-MS files of the three experimental groups with slightly different settings. We added all information in the Methods section paragraph "Liquid chromatography-mass spectrometry (LC-MS)". We included all analysis settings, as well as information on data processing. Of note, the new analysis does not affect any previously made data interpretation or scientific statements of the manuscript, however, as the re-analysis included 82 more proteins than initially, protein numbers and percentages are modified marginally in the revised manuscript.

Major concerns

1. The authors must provide much more clarity around how the mass spectrometry (MS) data was handled. For the peptide level data and include which library was used to search the data, and which modification were allowed etc. For the step where the peptide level data is used to infer the protein changes, did the authors sum /include common peptide which have the same amino acid sequence to more than one protein as oppose to inclusion of only proteotypic peptides; were isoforms only assigned if a peptide has an unique amino acid sequence for that isoform etc. and how cutoff for FDR and log₁₀ FC (note log₂FC is normally used) and whether FDR /p value was adjusted for multiple variables. In addition, how did the authors deal with

missing data such as if a protein seen in only subset of tissue or myocytes or changed in only subset of these animals/cultures. Please expand methods and include all this information including what library was used to search the data, if modification were allowed etc.

The Methods section was expanded extensively now providing all information on analysis and data processing.

2. The group must report all peptide and protein level data from their mass spectrometry for each animal and experiment. Coupled with lack of detail on how the data was processed makes it impossible to judge the quality of the data and thus, interpretation is correct. This data can go in as set of online supplement tables. The MS data need or should be uploaded to an open-source data repository like MASSIVE or Pride.

We agree with the reviewer. We have described the information about the data processing in the method section of the manuscript in detail and attached the table report of all peptide and protein level data in the supplements (Tab. S1). As stated in the data availability section all mass spectrometry data have been deposited to the ProteomeXchange Consortium via the PRIDE partner repository³, consisting of the MS instrument raw output files and the results of peptide/protein identification (mzIdentML). The project identifier is PXD058165 with token: Q6azH0UauMAk.

3. In figures S7, S8 and S9 figures of the total protein (SPL red) shows that for the tissue extracts the high MW cardiac proteins are missing (like myosin heavy chain, myosin binding protein C and titin). This is suggestive of incomplete solubilization or precipitation of the proteins. Without the MS data to support the work, one cannot check what maybe going on, as these dominant proteins must be present if full extraction or solubilization has occurred. It is also interesting /concerning that the total protein stain of the gel of the cardiomyocytes shown in S10 do have these dominant high MW proteins. Without knowing if full extract (and complete digestion) of the tissue has occurred then again, it is not possible ensure that interpretation is correct.

In order to rule out incomplete solubilization or precipitation we first checked for high-molecular-weight-proteins (high-MW proteins) in our LC-MS results. As shown in Resp. Fig. 2.1, the identified proteins are nicely distributed across a molecular weight ranging from 5.3 kDa to 3904.1 kDa with a median of 37.8 kDa. 21 proteins have a higher molecular weight than 300 kDa, including titin. The protein abundances of myosin heavy chain (MYH6), titin (TTN), and myosin binding protein C (MYBPC3) could be identified in all samples with equal trends, indicating proper solubilization of high-MW proteins.

Response Figure 2.1
Resp. Fig. 2.1: Distribution of molecular weight (MW) in kDa of 1878 identified proteins by LC-MS.

Second, we separated protein lysates of whole mouse LV tissue (LV) and adult murine cardiomyocytes (Murine CM) isolated with Urea/TEAB buffer, as done for proteome analysis. SPL red (PR926, NH DyeAGNOSTICS) was used for detection of total proteins directly after 12% SDS-PAGE (In-gel detection) and after blotting to a PVDF membrane (On-membrane detection). Immunoblotting with an antibody against MYBPC3 was performed (predicted 140kDa; #AP12436a; abcepta). In accordance with our proteome data, the high-MW-protein MYBPC3, could be detected in UREA/TEAB lysates of both LV mouse tissue and murine cardiomyocytes (Resp. Fig. 2.2). Of note, after blotting to PVDF membrane, the fluorescence signal of high-MW proteins was much weaker compared to the in-gel signal. This phenomenon was even more pronounced in LV mouse tissue lysates compared to murine cardiomyocytes (Resp. Fig. 2.2), which might explain the missing high-MW proteins in the supplemental figures. However, normalization of immunoblot data was done in a standardized manner with the LabImage software tool determining the intensity of proteins in a MW-range between 25 and 120 kDa.

Response Figure 2.2
Resp. Fig. 2.2: Detection of high-MW proteins in left ventricular mouse tissue (LV tissue) and murine cardiomyocytes (Murine CM) isolated with UREA/TEAB buffer. Total proteins were detected by fluorescence signal after staining with SPL red before (In-gel-detection) or after blotting (On-membrane detection) on PVDF membrane. The membrane was stained with antibodies against the myosin binding protein C (MYBPC3).

*Minor but important**1. Clearly state sex of animals (I assume they are all male).*

All mice were male in the initial manuscript. We now investigated a female cohort exposed to HFD+L-NAME and NO₂-OA. The data are included in the revised manuscript as Fig. S5a, b.

2. For figure 1A it may be helpful to include when NO₂-OA/OA was given and when samples were harvested.

As shown in Figure 1a and Suppl Fig. S1a, male mice receiving high fat diet and eNOS inhibition by L-NAME via the drinking water for 11 weeks were randomly divided into either two or three groups, dependent on the cohort. One group of mice was treated with NO₂-OA (HFD+L-NAME NO₂-OA), one with OA (HFD+L-NAME OA) and one group of mice was treated with vehicle (HFD+L-NAME vehicle) via osmotic pumps subcutaneously for additional 4 weeks until tissue was harvested after 15 weeks.

3. For figure 3. It is important add protein names for heat maps. It is also essential to include a table of all differentially expressed proteins for each comparison with their log change and adjusted FDR/p value.

We added the protein names and the mean z-scores for all heat maps in Tab. S3. Tab. S2 displays all proteins included in the analysis, the log changes and $-\log_{10}(\text{p-value})$ for each comparison indicating differentially expressed proteins.

Reviewer #3 (Remarks to the Author):

The authors investigated the effects of nitro-oleic acid therapy in a popular murine model of cardiometabolic HFpEF. The authors demonstrate a cardiac phenotype of HFpEF following 11 weeks of L-NAME + high fat diet feeding to the mice. The authors provide some evidence that nitro-oleic acid therapy attenuates diastolic dysfunction in the 2-hit HFpEF model. The study addresses a significant clinical issue and provides some data to demonstrate the potential benefit of a novel therapeutic agent. The authors provide extensive data to show that NO₂-OA improves cardiac mitochondrial function and overall metabolic function.

Overall, the studies have been performed in a careful manner and the data do support the author's conclusion that NO₂-OA exerts beneficial actions in this preclinical HFpEF model. We thank the reviewer for the affirming review.

I have a few comments and concerns regarding the paper.

(1) The authors provide data that NO₂-OA improves LV diastolic function. The entire study rests on 2-D ECHO data. The authors should provide data for exercise capacity (treadmill exercise) since this is essentially the most important outcome in patients that suffer from HFpEF in addition to overall quality of life. ECHO data alone is not sufficient.

As suggested by the reviewer, we performed treadmill experiments and calculated the work rate that was achieved (which considers the distance, animal's body weight, inclination of the treadmill and time). Vehicle-treated HFpEF mice achieved a significantly smaller work rate compared to control mice, whereas no significant difference was observed between control mice and NO₂-OA-treated HFpEF mice (Response Fig. 3.1a, Manuscript Fig. 1i). In addition, from the experimental sets that were investigated during the revision, we performed the 2-D ECHO investigation with a more extensive workflow and in a very sophisticated manner. Hereby, we could not only reproduce our previous findings (Resp. Fig. 1.1, Reviewer #1, note that initial data are derived from Charles River mice, revised data from Janvier Labs mice with similar results), but could also detect significant differences in left atrial (LA) area, which strongly corroborates the HFpEF phenotype and the impact of the treatment (Resp. Fig. 3.1b,

MS Fig. 1j). This was further supported by signs of pulmonary arterial remodeling in HFpEF mice, which was diminished by NO₂-OA treatment (Resp. Fig. 3.1c, d, MS Fig. 1k). Finally, invasive pressure volume analyses confirmed the augmentation in LV stiffness, which was attenuated by NO₂-OA (Resp. Fig. 3.1e, MS Fig. 1g, h).

Response Figure 3.1

Resp. Fig. 3.1: a) Work rate received from graded exercise test (N=18/14/17). b, c) End-systolic left-atrial area (LA area, s) and pulmonary vascular resistance derived from 2-D ultrasound (N=19/14/18). d) Representative images and quantification of muscularized pulmonary arteries (N=9/8/9). e) Left-ventricular end-diastolic pressure (LV EDP) and chamber stiffness index β_w were derived from hemodynamic measurements (N=11/7/6). Statistical significance was calculated by One-way ANOVA followed by Bonferroni's post-hoc test. p: *<0.05, **<0.01, ***<0.001, ****<0.0001.

(2) Data for HF severity biomarkers and for inflammation and oxidative stress should be provided. The authors should evaluate BNP and hsCRP, etc.

We thank the reviewer for this important suggestion. BNP was analyzed by immunoblot in isolated cardiomyocytes, which was not significantly but numerically increased in vehicle-, but not in NO₂-OA-treated HFpEF mice (Resp. Fig. 3.2a). Parameters reflecting fibrosis, cardiomyocyte hypertrophy (Resp. Fig. 3.2b, MS Fig. S2a, b) and left ventricular hypertrophy (MS Fig. 1b) show significant increases in HFD+L-NAME mice with no or subtle therapeutic effects of NO₂-OA.

Response Figure 3.2

Resp. Fig. 3.2: a) Protein level of brain-natriuretic peptide (BNP) was detected in isolated murine cardiomyocytes of chow, vehicle-treated HFD+L-NAME and NO₂-OA-treated HFD+L-NAME mice. The relative BNP level significantly negatively correlated with early diastolic mitral annulus velocity (e') and pulmonary artery volume time integral (PA VTI). Statistical significance was calculated by One-way ANOVA followed by Bonferroni's post-hoc test and Pearson's correlation ($N=8/6/5$). b) Representative images (scale bar 200 μ m) of LV sections stained with picrosirius red and analysis of fibrosis grades ($N=10/10/7$). Images were scored based on the amount of collagen deposition. Mean cell diameter of isolated cardiomyocytes assessed by pulse area analysis-based volumetry ($N=8/6/5$). Statistical significance was calculated by Fisher exact test for fibrosis. * presents analysis of chow vs. HFD+L-NAME vehicle and # analysis of HFD+L-NAME vehicle vs. NO₂-OA group. One-way ANOVA with Bonferroni's post-hoc test was used for cell diameter. $p: ***<0.001$.

To characterize inflammation, we quantified mRNA in LV tissue for several cytokines and measured plasma levels of myeloperoxidase (MPO) (Resp. Fig. 3.3a, b, MS Fig. S4a), which is a widely used marker for innate immune cell activation. MPO plasma levels were significantly increased in vehicle-treated HFpEF mice compared to the chow group, but not in NO₂-OA-treated HFpEF mice. However, the difference between vehicle and NO₂-OA was small, which is somewhat surprising in consideration of the well-known anti-inflammatory effects of NO₂-OA. Cytokine expression showed only marginal increases in HFpEF mice, without differences between vehicle and NO₂-OA treatment (Resp. Fig. 3.3b). According to that, we did not detect signs of oxidative damage in LV tissue reflected by unchanged levels of hyperoxidized peroxiredoxin in LV tissue (Resp. Fig. 3.3d, MS Fig. S4c), which might be explained by upregulation of the antioxidative defense. Superoxide dismutase (SOD1) was significantly upregulated in LV tissue of HFD+L-NAME mice compared to chow animals, which could be

normalized with NO₂-OA (Resp. Fig. 3.3c, MS Fig. S4d). Together, these data suggest that despite of systemic inflammation, in LV myocardium inflammatory activity is rather low and thus is unlikely to be causally related to diastolic dysfunction. We commented on this issue in the revised discussion section (ll. 720-727).

Response Figure 3.3

Resp. Fig. 3.3: a) Plasma level of myeloperoxidase (MPO) (N=10/10/6). b) mRNA level of inflammatory biomarkers: interleukin-1 β (IL1 β), tumor necrosis factor α (TNF α) and interleukin-6 (IL6) (N=10/10/11). c) mRNA level of superoxidase 1 (SOD1) and superoxidase 2 (SOD2) (N=9/10/10). d) Representative immunoblot and quantification of hyperoxidized peroxiredoxin (Prdx-SO_{2/3}) in left-ventricular mouse tissue (N=8/6/5). Statistical significance was calculated by One-way ANOVA followed by Bonferroni's post-hoc test; except for b Kruskal-Wallis test followed by Dunn's post-hoc test was used. p: *<0.05, **<0.01.

(3) The authors report on HFpEF in male mice only. The authors should also evaluate NO₂-OA for efficacy only in female mice.

To consider potential sex differences, female wild-type mice (8 weeks old, C57BL/6N, N=8) were exposed to normal chow diet or HFD+L-NAME (0.5 g/L in drinking water). After 15 weeks, heart function was analysed by echocardiography to assess baseline characteristics (Resp. Fig. 3.4a, MS Fig. S5a). Female HFD+L-NAME mice developed a milder phenotype compared to male HFD+L-NAME mice with significantly increased LV hypertrophy and LA area, but no change in E/e' ratio. Interestingly, female HFD+L-NAME mice did not significantly gain body weight but showed increased levels of visceral fat. After 15 weeks, all female HFD+L-NAME mice were treated with NO₂-OA (oral administration daily, N=7) for 4 weeks. Echocardiography revealed significantly improved diastolic function shown by E/e' ratio and LA area in female HFD+L-NAME mice after treatment with NO₂-OA compared to baseline (Resp. Fig. 3.4b, MS Fig. S5b), confirming the efficiency of NO₂-OA in metabolic cardiomyopathy in female mice.

Response Figure 3.4

Resp. Fig. 3.4: a) Female mice were exposed to normal chow diet or HFD + L-NAME. After 15 weeks heart function was evaluated by echocardiography (Baseline) (N=8/8). b) All HFD+L-NAME mice were treated with NO₂-OA for 4 weeks and echocardiography was performed at the final timepoint. One mouse refused oral administration of NO₂-OA (N=7/7). Statistical significance was calculated by unpaired Student's t-test for a and paired Student's t-test for b. p: *<0.05, ***<0.001.

(4) The authors should provide data for left ventricular pressures and for LV wall thickness and chamber dimensions.

We performed pressure-volume loops, which revealed significantly increased LV end-diastolic pressure in vehicle- but not NO₂-OA-treated HFD+L-NAME mice compared to chow animals (Resp. Fig. 3.5a, MS. Fig. 1g). End-systolic pressure was enhanced in HFpEF mice with a more pronounced increase in NO₂-OA-treated HFD+L-NAME mice (Resp. Fig. 3.5a). Left ventricular posterior wall thickness as assessed by echocardiography showed increased LV hypertrophy in HFD+L-NAME mice compared to chow animals, which was less pronounced in NO₂-OA-treated HFD+L-NAME mice (Resp. Fig. 3.5b, MS Fig. 1b).

Response Figure 3.5

Resp. Fig. 3.5: a) Left-ventricular end-diastolic pressure (LV EDP) and left-ventricular end-systolic pressure (LV ESP) gathered from pressure-volume loop analysis (N=11/7/6). b) Left-ventricular posterior wall thickness (LVPW)

and end-diastolic volume (EDV) taken from cardiac ultrasound (N=19/14/18). Statistical significance was calculated by One-way ANOVA followed by Bonferroni's post-hoc test. p: * <0.05 , *** <0.001 .

(5) One of the major discrepancies in the HFpEF field is the report by Schiattarella and Hill that the primary driver of HFpEF in this same model is nitrosative stress resulting from activation of iNOS and overproduction of NO. This is in sharp contrast to preclinical and clinical reports of attenuated NO bioavailability and impaired endothelial dysfunction leading to coronary microvascular dysfunction and worsening HFpEF.

The authors must evaluate how NO₂-OA impacts nitrosative stress. NO₂-OA should increase NO bioavailability and the authors should measure cardiac and global NO levels in HFpEF Control and HFpEF + NO₂-OA mouse groups.

We absolutely agree with the reviewer, that the finding of Schiattarella et al. on nitrosative stress in this model needs to be considered. We performed real time qRT-PCR analyses for mRNA quantification of NOS2 and XBP1s in LV tissue using the same primer sequences as Schiattarella et al. Surprisingly, neither after 5 weeks, nor after 15 weeks of HFD+L-NAME we could detect upregulated NOS2 and XBP1s expression (Resp. Fig. 3.6a). In accordance, we did not detect increased levels of 3-nitro-tyrosine, an established marker of nitrosative stress (Resp. Fig. 3.6b). Given that nitrosative stress is related to inflammatory processes, this discrepancy with Schiattarella's findings might be due to the hygienic environment and thus different immune activities across different animal facilities, which is becoming more and more evident in different laboratories. There are currently efforts underway to investigate this issue in multicenter animal studies.

We furthermore quantified the phosphorylation (Ser239, pVASP) of the vasodilator-stimulated phosphoprotein (VASP), which is an established surrogate of NO-bioavailability. In myocardial tissue we were not able to receive pVASP signals, so we used snap-frozen aortic tissue, which should reflect coronary NO levels. The amount of pVASP in aortic tissue was significantly reduced in both vehicle- and NO₂-OA-treated HFpEF mice (Resp. Fig. 3.6c, MS. Fig. S4b). We conclude from these results, that eNOS inhibition by L-NAME is mainly responsible for reduced NO-bioavailability, which cannot be reversed by NO₂-OA. Both iNOS activation and oxidative stress are absent or mild in our model, so that both do not influence NO-bioavailability in a relevant manner. These findings are supported by enhanced peripheral vascular resistance in both HFpEF groups compared to chow animals, as reflected by arterial elastance (E_a) derived from pressure volume measurements (Resp. Fig. 3.6d, MS Fig. 1I). While these data are unexpected, they support our hypothesis that other than these mechanisms account for an improvement of diastolic LV function in our model.

Response Figure 3.6

• Chow ▼ HFD+L-NAME vehicle ◆ HFD+L-NAME NO₂-OA

Resp. Fig. 3.6: a) mRNA level of inducible nitric oxide synthase (NOS2) and X-box-binding protein 1 (Xbp1b) (5wk: N=8/9/9; 15wk: N=9/10/10). b) Representative immunoblot and quantification of 3-nitro-tyrosine (NO₂-Tyr) (N=8/6/5). c) Level of phosphorylation (Ser239, pVASP) of the vasodilator-stimulated phosphoprotein (VASP) was quantified via immunoblot (N=10/6/6) in aorta. d) Arterial elastance (Ea) was derived from pressure volume measurements (N=11/7/6). Statistical significance was calculated by One-way ANOVA followed by Bonferroni's post-hoc test. p: *<0.05.

The authors should also measure blood pressure.

Unfortunately, we do not have the equipment for blood pressure measurements in conscious mice available. However, from invasive pressure volume analyses we conclude that blood pressure is unlikely to be reduced by NO₂-OA treatment (Res. Fig. 3.5a), which might be explained by similar NO-bioavailability due to eNOS inhibition.

This is a critically important issue that needs to be addressed.

(6) The authors fail to fully address the limitations of this preclinical model in which they induce obesity, hypertension, and HFpEF in an extremely short time. There are several other weakness to this model.

We rephrased the relevant section and added limitations of the mouse model.

References:

1. Burkhoff D, Mirsky I, Suga H. Assessment of systolic and diastolic ventricular properties via pressure-volume analysis: a guide for clinical, translational, and basic researchers. *Am J Physiol Heart Circ Physiol*. 2005;289:H501-512. doi: 10.1152/ajpheart.00138.2005
2. Rath S, Sharma R, Gupta R, Ast T, Chan C, Durham TJ, Goodman RP, Grabarek Z, Haas ME, Hung WHW, et al. MitoCarta3.0: an updated mitochondrial proteome now with sub-organelle localization and pathway annotations. *Nucleic Acids Res*. 2021;49:D1541-D1547. doi: 10.1093/nar/gkaa1011
3. Perez-Riverol Y, Bai J, Bandla C, Garcia-Seisdedos D, Hewapathirana S, Kamatchinathan S, Kundu DJ, Prakash A, Frericks-Zipper A, Eisenacher M, et al. The PRIDE database resources in 2022: a hub for mass spectrometry-based proteomics evidences. *Nucleic Acids Res*. 2022;50:D543-D552. doi: 10.1093/nar/gkab1038

Response to Reviewer

Reviewer #2:

Many of my concerns have been addressed however there are still some challenges remaining.

Major issues

1. MS methods have been expanded to include some of the needed information however there remains some important clarifications needed.

To confirm the integrity of our proteome analysis, we thoroughly checked the raw data and analysis methods with our MS expert Thomas Patschkowski, PhD.

Please clarify the following.

• *“only unique peptides were included in the analysis” means that peptides with amino acid sequence that can be assigned to a single unique protein (prototypic peptide) was used. This may not be the case (see point 2). As well confirm that only the prototypic peptides were used in the protein identification and quantification.*

We confirm that only proteotypic peptides, which are denominated as “unique peptides” in the *Proteome Discoverer* software, were employed for protein identification and quantification. Please find further information in Response Figure 1.

Response Figure 1

Screenshots demonstrating the settings in *Proteome Discoverer* software

Quantification Node

Precursor Ions Quantifier 11

1. General Quantification Settings
Peptides to Use Unique

Peptides to Use
Specifies which peptides are used for quantification.
Unique: Only peptides that are not shared between different proteins or protein groups are used for the protein quantification.

2. Precursor Quantification
Precursor Abundance Based On Intensity

Protein Grouping Node

Protein Grouping 5

1. Protein Grouping
Apply strict parsimony principle True

Explanations:

The settings in the “**Precursor Ions Quantifier**” Node ensure that only unique (proteotypic) peptides are used for quantification.

The setting in “**2. Precursor Quantification**” uses Precursor Intensity as quantification method.

As the “**Apply Strict Parsimony Principle**” parameter of the Protein Grouping node is set to “**True**”, the application removes all protein groups that have no unique peptides among the peptides that it considers for the protein grouping process.

Figure 1: Settings in *Proteome Discoverer* used for protein identification and explanation.

- *“Only proteins with valid values in all replicates of each group were considered” means the protein had to be present in all biological replicate in all sample or do you mean in each experimental group?*

We excluded a protein from the analysis, if it was not present in all samples (i.e. biological replicates = 23 animals) of all three experimental groups. For better clarity, we rephrased the sentence in the method section (see line 290). We added this issue to the limitations section, as it is a very strict selection approach.

- *“Proteins considered as potential contaminants were filtered out” means those that were from blood (e.g., hemoglobin?). Be specific.*

We used the database for contaminants included in the Proteome Discoverer software (PD_Contaminants database v2015_5). In short, the database includes frequent contaminants, such as keratins, bovine serum albumin and trypsin, from different species, such as *bos taurus*, *homo sapiens* and *mus musculus*, to identify obvious sample contaminations. For clarity we excluded the contaminants in table S1.

- *The authors set the FDR at peptide and protein level at 0.05, which is high (typically 0.01). Please add this to the limitation section as it is due, in part, to the MS instrument used.*

We thank the reviewer for this important note, which pointed out a mistake in our method section. The minimum false discovery rate (FDR) at which a peptide-spectrum match (PSM) was considered significant was indeed set to 0.01. This becomes obvious by the quality q-values being less than 0.01 for each identified peptide or peptide group (see Table S1). However, for post hoc analysis, the protein abundances of each protein were processed with the Perseus software tool 2.0.9.0 (MPI Biochemistry). Here, two experimental groups (HFD+L-NAME vehicle vs. chow; HFD+L-NAME NO₂-OA vs. HFD+L-NAME vehicle) were compared and differentially abundant proteins were defined as those with a fold change > 1.5 and FDR < 0.05 (Manuscript Fig. 2b and 2d). For mouse studies an FDR of 0.05 for the identification of statistically significant differences on protein level might be common¹. We rephrased the method section accordingly (line 276-294).

2A. Very importantly, the authors have NOT included the peptide level data (both sequence and modified residues are needed) making it impossible to determine if the protein identification and quantification are correct.

We apologize for not having indicated where to find the peptide level data. These can be found in the mzIdentML document, which we have deposited to the ProteomeXchange Consortium via the PRIDE partner repository. In addition, Table S1 contains peptide level data including sequence and modified residues. For more clarity, we modified Table S1 and explained the columns in a table legend.

	Label	Description
Protein level	Exp. q-value: Combined	q-values derived from the validation for the proteins and protein groups. An experimental q-value: combined of 0 indicates that the false discovery rate (FDR) is effectively zero for the combined analysis of a specific protein or peptide. This means that the identification is highly confident, with no expected false positives among those meeting this threshold.
	Sum PEP Score	Sum of posterior error probability (PEP) score; protein score calculated as the sum of the negative logarithms of the PEP values of all the connected peptide-spectrum matches (PSMs).
	Coverage [%]	Percent sequence coverage calculated by dividing the number of amino acids in all found peptides by the total number of amino acids in the entire protein sequence.
	# Peptides	Number of peptides (unique and razor peptides).
	# PSMs	Peptide-spectrum match (PSM) for annotated peptides of this protein.
	# Unique Peptides	Number of unique peptides that are contained in only one protein group.

	# AAs	Number of amino acids.
	MW [kDa]	Molecular weight of the protein in kilo-daltons.
	calc. pI	Calculated isoelectric point of the protein.
	Abundance	Quantitative value calculated for the identified protein for each sample.
Peptide level	Annotated Sequence	Annotated peptide sequence.
	Modifications	Name of the posttranslational modification and amino acid position in the annotated peptide sequence.
	Quality PEP	Posterior error probability (PEP) for the identified peptide group.
	Quality q-value	q-value for the identified peptide group.
	# Protein Groups	Number of identified protein groups based on the annotated peptide sequence.
	# Proteins	Number of identified proteins based on the annotated peptide sequence.
	# PSMs	Peptide-spectrum match (PSM) for the annotated peptide sequence.
	Master Protein Accessions	List of UniProt accession numbers of each master protein for each protein group of the annotated peptide sequence.
	Positions in Master Proteins	Position of the annotated peptide in each identified master protein.
	# Missed Cleavages	The number of missed cleavages in the annotated peptide.
	Theo. MH+ [Da]	The theoretical MH+ mass of the modified peptide, in daltons.
	XCorr (by Search Engine): A3 Sequest HT	Score calculated by the Sequest HT search engine for peptide matches; the search engine provides peptide-spectrum matches (PSMs) that have the best XCorr score for each spectrum.
	Sequence in Protein	Sequence of annotated peptide in the protein.
	Positions in Proteins	Position of the annotated peptide in the protein.

Peptide level data are found in the grouped excel table (Table S1) as indicated in Response Figure 2.

Response Figure 2

Screenshot demonstrating the peptide level in table S1.

UniProt accessi numb	UniProt prote name	Protein name	Exp. q-value: Combined	Sum P ^{score}	Coverage [%]	# Pepti	# PSA	# Uni Pepti	# AAs	Molecular weight [kDa]	
P09542	MYL3_MOUSE	Myosin light chain 3		0	1077,521	93	38	11542	35	204	22,4
		Annotated Sequence	Modifications	Quality PEP	Quality q-value	# Protein Groups	# Proteins	# PSMs	Master Protein Accessions	Positions in Master Proteins	
		[R].VLGHFKQEELNSK.[M]		7,07E-11	3,85E-05	1	1	126	P09542	P09542 [104-116]	
		[K].EAEFDASK.[I]		1,76E-08	1,51E-04	1	1	83	P09542	P09542 [43-50]	
		[R].LTDVEEK.[L]		5,03E-03	2,87E-04	1	1	106	P09542	P09542 [173-180]	
		[R].HVLATLGER.[L]		1,10E-04	3,85E-05	1	1	595	P09542	P09542 [164-172]	
		[K].EGNGTVMGAEIR.[H]	1xOxidation [M7]	1,11E-05	3,85E-05	3	3	316	P09542; P05977; Q60605	P09542 [152-163]; P05977 [136-147]; Q60605 [99-110]	
		[K].EAEFDASK.[I]		9,02E-04	1,17E-04	1	1	65	P09542	P09542 [43-52]	
		[R].VFDKNGTVMGAEIR.[H]	1xOxidation [M11]	2,86E-14	3,85E-05	3	3	1232	P09542; P05977; Q60605	P09542 [148-163]; P05977 [132-147]; Q60605 [95-110]	
		[K].AAPKAAAPAPAAAPAAAPAAPEPERPK.[E]		1,08E-16	3,85E-05	1	1	6	P09542	P09542 [15-42]	
		[K].EGNGTVMGAEIR.[H]		3,40E-05	3,85E-05	3	3	78	P09542; P05977; Q60605	P09542 [152-163]; P05977 [136-147]; Q60605 [99-110]	
		[K].AAPAPAAAPAAAPAAPEPERPK.[E]		1,25E-17	3,85E-05	1	1	112	P09542	P09542 [20-42]	
		[K].IYTGCGDVLV.[A]	1xCarbamidomethyl [C6]	1,81E-04	3,85E-05	1	2	229	P09542	P09542 [80-90]	
		[R].ALGQWPTQAEVLR.[R]		6,31E-07	3,85E-05	1	1	294	P09542	P09542 [91-103]	
		[K].AAPKAAAPAPAAAPAAAPAAPEPERPKAEAFDASK.[I]		1,40E-16	3,85E-05	1	1	29	P09542	P09542 [15-50]	
		[R].VFDKNGTVMGAEIR.[H]		1,26E-12	3,85E-05	3	3	588	P09542; P05977; Q60605	P09542 [148-163]; P05977 [132-147]; Q60605 [95-110]	
		[K].AAPAPAAAPAAAPAAPEPERPKAEAFDASK.[I]		2,03E-17	3,85E-05	1	1	168	P09542	P09542 [20-50]	
		[K].GEMKITYGCGDVLV.[A]	1xCarbamidomethyl [C10]	7,84E-09	3,85E-05	1	1	27	P09542	P09542 [76-90]	
		[K].AAPAPAAAPAAAPAAPEPERPKAEAFDASK.[I]		6,75E-17	3,85E-05	1	1	79	P09542	P09542 [20-52]	
		[R].HVLATLGERLTDVEEK.[L]		1,21E-15	3,85E-05	1	1	83	P09542	P09542 [164-180]	
		[K].EAEFLFDRTPKGEEMK.[I]		8,00E-05	3,85E-05	1	1	4	P09542	P09542 [65-79]	
		[K].NKDTGTEDYFVEGLR.[V]		1,39E-11	3,85E-05	1	1	431	P09542	P09542 [133-147]	
		[K].EAEFLFDRTPK.[I]		3,54E-05	3,85E-05	1	1	19	P09542	P09542 [65-75]	

Explanations:

For each protein (e.g. cardiac myosin light chain 3, MYL3, P09542) the annotated peptides and their identified posttranslational modifications can be found as subpoints under the “+” symbol. Unique peptides are indicated as peptides with only one protein group and master protein accession number. For annotated peptides with more than one protein group all master protein accession numbers are displayed.

Example:

The myosin light chain 3 (MYL3, P09542) was identified by 35 unique peptides (orange background). 38 peptides were matched to MYL3 in total (blue background). The annotated peptide sequence “[K].EGNGTVMGAEIR.[H]” was matched to three protein groups with the master proteins (cardiac myosin light chain 3, MYL3, **P09542**; skeletal muscle isoform myosin light chain 1/3, MYL1, **P05977**; myosin light chain polypeptide 6, MYL6, **Q60605**) (red arrow). This peptide is not annotated as unique peptide and was not used for quantification.

Figure 2: Peptide level in Table S1.

Data for example in supplement table 1 (485599_1_data_set_9946709_sncr6n as well as in other tables such as 485599_1_data_set_9946716-sncr93) suggests there is an issue with protein identification being exclusively based on proteotypic peptides is illustrated by alpha-actin isoform assignments. Alpha-actin is extremely highly conserved (>95%) between the three isoforms expressed in different muscle types. In table 1, only 2 isoforms of alpha actin isoforms are listed and neither is the cardiac isoform, the isoform that should be present. The data reported in Table 1 for alpha-actin is i) uniprot ID P68134, gene name Acta1 (protein name is ACTS) which is actin, alpha skeletal muscle and ii) uniprot ID P62737, gene name Acta2 (protein name ACTA) which is actin, aortic smooth muscle. The actin isoform that should be in cardiac muscle is uniprot ID P68033 gene name Actc1 (protein name ACTC) which is actin alpha cardiac muscle. As alpha actin is highly conserved, this success that incorrect identification has occurred. this maybe larger issues a P05977 myosin light chain 1/3 skeletal muscle while cardiac is myosin light chain 2 and 3. The authors need to ensure that truly prototypic peptides are being used for identification and quantification.

We thoroughly inspected the data from MS analysis and are confident, that issues of incorrect identification have not occurred. We set the Proteome Discoverer software to include only unique peptides for protein identification and to apply a strict parsimony principle (see Response Figure 1). We agree that cardiac actin (uniprot ID P68033, protein name ACTC) is not included in Table S1. The reason is that no unique peptide was identified for this protein. If we omit this setting from the software, ACTC is listed, but with zero unique peptides (Response Figure 3). An alignment of the three actin isoforms P62737, P68134 and P68033 reveals, that the only unique peptide of P68033 is located in

the N-terminus of the protein. The N-termini of the proteins P62737, P68134 and P68033 could not be identified in our analysis. Therefore, an unambiguous identification of P68033 was not possible (Response Figure 3).

Response Figure 3

Screenshot from *Proteome Discoverer* demonstrating that the proteins P62737, P68134 and P68033 can be identified, if the “Apply Strict Parsimony Principle” parameter is set to “False”.

	Accession	Description	Exp. q-value	Sum PEP Score	Coverage [%]	# Peptides	# PSMs
1	P68033	Actin, alpha cardiac muscle 1 [OS=Mus musculus]	0.000	1695.049	94%	55	2033
2	P62737	Actin, aortic smooth muscle [OS=Mus musculus]	0.000	1644.803	94%	53	1959
3	P68134	Actin, alpha skeletal muscle [OS=Mus musculus]	0.000	1415.241	88%	52	1796

Alignment of the N-termini of proteins P62737, P68134 and P68033

sp|P62737|ACTA_MOUSE MCEEEDSTALVCDNGSGLCKAGFAGDDAPRAVFPSIVGRPRHQGVMVGMGQKDSYVGDEA 60

sp|P68134|ACTS_MOUSE MCDEDETTALVCDNGSGLVKAGFAGDDAPRAVFPSIVGRPRHQGVMVGMGQKDSYVGDEA 60

sp|P68033|ACTC_MOUSE **MCDDEETALVCDNGSGLVK**AGFAGDDAPRAVFPSIVGRPRHQGVMVGMGQKDSYVGDEA 60

:::***

Protein Identification Details

Coverage ProteinCard

Actin, alpha cardiac muscle 1 [OS=Mus musculus]

Annotate PTMs reported in Uniprot
 Show only PTMs
 Include PSMs that are Filtered Out

Coverage: **94.16%**

Found Modifications:

Modification	1	11	21	31	41	51	61	71	81	91
Modifications P68033	MCDDEETAL	VCDNGSGLVK	AGFAGDDAPR	AVFPSIVGRP	RHQGVMVGMG	QRDSYVGDEA	QSKRGILTLLK	YPIERGIITN	WDDMEKIMHH	TFYNELRVAP
Modifications P68033	EEHPTLLTEA	PLNPKANREK	MTQIMFETEN	VPAMVVAIQA	VLSLYASGRY	TGIVLDSGEG	VTHNVPPIYEG	YALPHAIMRL	DLAGRDLYDY	LMKILTERGY
Modifications P68033	SFVTTAEREI	VRDIKEKLCY	VALDFENEMA	TAASSSSLEK	SYELPQQQVI	TIGNERFRCP	ETLFPQSFIC	MESAGIHETT	YNSIMKCDID	IRKDLYANNV
Modifications P68033	LSGOTTMYPG	IADRMQKEIT	ALAPSTMKIK	IIAPPERKYS	VMIGGSILAS	LSTFOOMGIS	MREYDEAGPS	IVHRKCF		

Figure 3: Actin alpha cardiac muscle 1 can be identified using razor peptides (above). Alignment of the three actin isoforms reveals high sequence homology (down).

The cardiac myosin light chain isoforms 2 (P51667, MLRV) and 3 (P09542, MYL3) have been identified (see Table S1, line 4758 and 4385 and Response Figure 4).

Response Figure 4

Figure 4: Profile Plot showing the identification of the actin isoforms and cardiac myosin light chains in every sample. A relative intensity value is plotted on the y-axis.

2B. The authors have provided the requested protein data.

• Supplement table 1. Remove columns A, B and C (e.g., master protein column) or define their meaning, add in Uniprot identifier as these define the protein, add in the protein name (not just the gene name), define each column labeled as peptide (e.g., column J is labeled # peptides includes those peptides that are shared between more than one protein as well as those that are unique to a single protein (prototypic peptides). How column J differs from the number of peptides listed in columns R and S is not clear. In the methods, the authors stated only unique proteolytic peptides (column L) are used for protein. Therefore, consider removing the other columns. As well, confirm what are the values listed for each sample (is this average or total sum of the peak area or of the spectral counts for each proteotypic peptide. How these values are used for the protein quantification in the other tables must be clarified including if the quantification is based on unique prototypic peptides. If not, these data need to be redone using quantitation only the prototypic peptides.

We thank the reviewer for the notion, that tables were unclear. We modified Table S1 and included a legend/description of every column for more clarity (see answer to 2A).

The protein abundance for every sample shown in Table S1 was calculated by the Proteome Discoverer software with the settings explained in response to comment 1 and Response Figure 1. Only unique peptides were used for protein quantification (abundances in Table S1). In detail, the Proteome Discoverer software calculates the raw quantification values and associates them with the peptide-spectrum matches (PSMs) identified by the Sequest HT search engine. The quantification

values are peak heights (intensity) for precursor ion quantification. The quantification channel values linked to the PSMs are summed to the peptide groups and proteins. PSMs that do not meet the criteria set by the method are not included in this sum. Otherwise, this methodology aggregates all contributing signals of the protein, irrespective of the charge and modification state of the peptides. The result is quantification channel abundance values for all channels from all files for the proteins. Abundances of proteins (Tables S1 columns M to AI) were used for further analysis with the Perseus software.

• *Supplement table 2. For each Tab, reduce decimal points to two or three spaces after the decimal place for -log P value and difference. Define what is meant as it is not clear how this was determined, define or remove columns F to I and define animal experiment group of each sample in columns M-AI.*

For more clarity table S2, S3 and S4 were included in the source data file. The data presented in this tables are visualized in Fig. 2b and 2d (Volcano plots), as well as in Fig. 3a, 4b, 6a and S7 (heat maps). The data were processed with the Perseus software tool version 2.0.9.0 (MPI Biochemistry) as indicated in the figure legends.

• *Supplement Table 3, 4 and 5 (protein and lipid). For each Tab, reduce decimal points to two-three spaces after the decimal place.*

The data represented in S2, S3 and S4 are now included in the source data file (see answer above). Supplemental Table S2 presents now the lipid data. The data are presented with two-three decimal spaces as recommended by the reviewer.

3. Thank you for adding in the information to the reviewers comments about high MW proteins for mass spectrometry (Figure 2.1) and the additional gel images (figure 2.2). These were very informative and must be included in the online supplement as this data confirms that there is an under representation in the quantity of the high MW proteins. Although detected by MS the protein ratio of MYHC6, MYBPC and titin the quantitation of these proteins do not seem to match up with respect to the lower MW proteins. This can be shown by plotting the dynamic range (total intensity of the prototypic peptides) across the proteome quantify and show where the representative high and low MW protein (e.g., cardiac TnT, cardiac TnI, MHY6, Titin, cardiac alpha actin).

This issue is better illustrated by the gel stained images which clearly shows low amount of staining of the high MW proteins especially compared to alpha cardiac actin (the dark 42K band (which is not identified by MS as discussed above)) in Figure 2.2 A and B added to the response to reviewers (note: it is not clear if C is overexposed as no linear range of the westerns was included). This under representation of the high MW protein is also seen in the gels for tissue (Figure 12.a, 13a, 14a) in the online supplement of the main manuscript (compare to the cardiomyocytes Figure 15) suggesting this is a wider spread issue. This needs to be addressed and should be included in a limitation section.

Response Figure 5 was included in the manuscript (Fig. S16). Methodological limitations of the proteome analysis were added to the limitation section.

Response Figure 5

Figure 5: Detection of high-molecular weight proteins in left ventricular mouse tissue (LV tissue) and murine cardiomyocytes (Murine CM) isolated with UREA/TEAB buffer. Total proteins were detected by fluorescence signal after staining with SPL red before (In-gel-detection) or after blotting (On-membrane detection) on PVDF membrane. The membrane was stained with antibodies against the myosin binding protein C (MYBPC3).

Minor issues

1. Thank you for including female animals.
2. Thank you for expanding the information.
3. Thank you for expanding the information.

References

1. Takasugi M, Nonaka Y, Takemura K, Yoshida Y, Stein F, Schwarz JJ, Adachi J, Satoh J, Ito S, Tomblin G, et al. An atlas of the aging mouse proteome reveals the features of age-related post-transcriptional dysregulation. *Nat Commun.* 2024;15:8520. doi: 10.1038/s41467-024-52845-x

Response to Reviewer

Reviewer #2 (Remarks to the Author):

Thank you for the careful responses to the remaining queries. The changes made in the manuscript and online supplements have helped with transparency and confidence. This remains an exciting and important study.

We are pleased with the positive assessment.

Reviewer #2 (Remarks on code availability):

I did not run it but did take a look through the app and it was fine. I did notice that the demo dataset had an issue "Sorry about that, but we can't show files that are this big right now." and I am running the most recent windows. But it could be something on my end. The read me files are fine and transparent.

The "demo dataset" is a single image used to test our software tool. Due to its file size, the image cannot be displayed directly in the browser, resulting in an error message. However, the image can be downloaded via the "Raw" button at the top right or the "View raw" button and then used in the software tool (Response Figure 1). This is not a bug in the code but a limitation of GitHub to save bandwidth.

Response Figure 1: Screenshot showing the error message. Blue arrows indicate the two buttons to download the image.